# Overlapping Multi-Domain Spectral Method for Conjugate Problems of Conduction and MHD Free Convection Flow of Nanofluids over Flat Plates

**Musawenkhosi Mkhatshwa [1],* , Sandile Motsa [1,2] and Precious Sibanda [1]**

1   School of Mathematics, Statistics and Computer Science, University of Kwazulu Natal, Scottsville, Pietermaritzburg 3209, South Africa

2   Department of Mathematics, Faculty of Science and Engineering, University of Eswatini, Private Bag 4, Kwaluseni, Eswatini

*   Correspondence: patsonmkhatshwa@gmail.com; Tel.: +268-7836-8742

**Abstract:** An efficient overlapping multi-domain spectral method is used in the analysis of conjugate problems of heat conduction in solid walls coupled with laminar magnetohydrodynamic (MHD) free convective boundary layer flow of copper (Cu) water and silver (Ag) water nanofluids over vertical and horizontal flat plates. The combined effects of heat generation and thermal radiation on the flow has been analyzed by imposing a magnetic field along the direction of the flow to control the motion of electrically conducting fluid in nanoscale systems. We have assumed that the nanoparticle volume fraction at the wall may be actively controlled. The dimensionless flow equations are solved numerically using an overlapping multi-domain bivariate spectral quasilinearisation method. The effects of relevant parameters on the fluid properties are shown graphically and discussed in detail. Furthermore, the variations of the skin friction coefficient, surface temperature and the rate of heat transfer are shown in graphs and tables. The findings show that the surface temperature is enhanced due to the presence of nanoparticles in the base fluid and the inclusion of the thermal radiation, heat generation and transverse magnetic field in the system. An increase in the nanoparticle volume fraction, heat generation, thermal radiation, and magnetic field parameter enhances the nanofluid velocity and temperature while reducing the heat transfer rate. The results also indicate that the Ag–water nanofluid has higher skin friction and surface temperature than the Cu–water nanofluid, while the opposite behaviour is observed in the case of the rate of heat transfer. The computed numerical results are compared with previously published results and found to be in good agreement.

**Keywords:** multi-domain overlapping technique; bivariate spectral quasilinearisation method; conjugate heat transfer; MHD free convection; radiation; heat generation; nanofluid; vertical and horizontal flat plates

## 1. Introduction

Conjugate heat transfer (CHT) is the interaction between the conduction and the buoyancy forced flow of fluid along a solid surface. In numerous applications, the effect of conduction within the solid wall is significant and thus must be taken into account. Such applications include heat exchangers, heaters, nuclear reactors, and pipe insulation systems. In these applications, the analysis of CHT mechanisms, the coupling of the conduction in the solid body and the convection in the fluid surrounding is important [1]. CHT problems, in which the coupled heat transfer processes between conduction and convection mechanisms are considered simultaneously, have been studied by several researchers in the case of Newtonian fluids. For example, Miyamoto et al. [2] reviewed the early theoretical and experimental work of conjugate free convection including the methods and

the principal results in the previously obtained solutions of conjugate problems. Miyamoto et al. [2] considered CHT problems of free convection from a vertical plate with a uniform temperature or a uniform heat flux on the outside surface of the plate. Sparrow and Chyu [3] studied CHT problems for a vertical fin with forced convection. Merkin and Pop [4] analyzed CHT over a vertical flat plate using an efficient finite-difference scheme. Pop et al. [5] presented a detailed numerical study of the conjugate mixed convection flow along a vertical flat plate. Luna et al. [6] investigated CHT across a thin horizontal wall separating two fluids at different temperatures numerically and asymptotically. Vasquez and Bula [7] studied the CHT process in cooling a horizontal plate in a steady state condition. Hajmohammadi and Nourazar [8] investigated conjugate forced convection heat transfer from a good conducting horizontal plate with temperature-dependent thermal conductivity. The horizontal plate was heated with uniform heat flux at the lower surface and cooled at the upper surface under laminar forced convection flow. Findings showed that, for a good conducting plate with a finite thickness, the distribution of the conjugate heat flux at the upper surface is significantly affected by the plate thickness. Yu and Lin [9] analyzed conjugate free convection over a vertical and horizontal plate using Keller's finite-difference method. They proposed the new conjugate parameters and novel dimensionless coordinates to solve the conjugate free convection problem on vertical and horizontal plates. Hsiao [10] analysed the conjugate problems of conduction in solid and free convection in fluid flow using a novel improved formula. The flow equations were solved numerically using the finite difference, Runge–Kutta and Shooting method.

Many studies have been performed on magnetic field and heat generation effects on magnetohydrodynamic (MHD)-conjugate heat transfer. Azim and Chowdhury [11] investigated MHD-conjugate free convection from an isothermal horizontal circular cylinder with Joule heating and heat generation in the presence of a magnetic field. Azim et al. [12] studied the problem of steady CHT through an electrically-conducting fluid for a vertical flat plate with a transverse uniform magnetic field. Kaya [13] investigated mixed convection heat transfer about a thin vertical plate with magneto and CHT effects in a porous medium. Kaya [14] studied the effect of CHT on MHD mixed convection about a vertical slender hollow cylinder. Mamun et al. [15] studied the effects of conduction and viscous dissipation on natural convection flow of an incompressible, viscous and electrically conducting fluid with a transverse magnetic field. Mamun et al. [16] investigated the magnetic field, viscous dissipation and heat generation effects on natural convection flow of incompressible, viscous and electrically conducting fluid along a vertical flat plate with conduction. Hosain and Azim [17] studied the effects of viscous dissipation and heat generation on MHD conjugate free convection flow from an isothermal horizontal circular cylinder when the magnetic field was applied.

In the studies mentioned above, the fluid was assumed to be regular. However, traditional fluids such as water, oil and ethylene glycol might not have enough thermal conductivity to provide the desired efficiency. A good way to overcome this limitation is to add some solid nanoparticles with high thermal conductivity to the fluid. The resulting fluid is a suspension of the solid nanoparticles in the base fluid, which is called nanofluid. The thermal conductivities of nanofluids are believed to be greater than those of the base fluid due to the high thermal conductivity of the nanoparticles. Numerous investigations have been done on the effect of nanoparticles on thermal performance. For example, Choi et al. [18] experimentally studied the effective thermal conductivity of a nano-solid-liquid mixture. Their results revealed that the dispersion of a small amount (<1% by volume) of carbon nanotubes in a liquid increases its thermal conductivity remarkably (nearly 200%). Nanofluids have many applications in heat transfer such as microelectronics, fuel cells, Pharmaceutical processes, Hybrid-powered engines, engine cooling vehicles, domestic refrigerator, heat exchanger, nuclear reactor coolant, space technology, and boiler flue gas temperature reduction. Nanoparticles can exist in a variety of types such as metals, metal oxides, carbides, and carbon. The most common types of nanofluids available commercially include Aluminium oxide ($Al_2O_3$), Titanium oxide ($TiO_2$), Copper (Cu), and Silver (Ag)–water nanofluid [19].

The study of CHT in nanofluids has attracted the interest of many researchers. Jafarian et al. [20] studied CHT in MHD mixed convective flows of nanofluid about a vertical slender hollow Cylinder embedded in a porous medium. Nimmagadda and Venkatasubbaiah [21] analyzed CHT in a micro-channel using novel hybrid nanofluids ($Al_2O_3 + Ag$/Water). Patrulescu and Grosan [22] studied CHT in a vertical channel filled with a nanofluid adjacent to a heat generating solid domain. Zahan and Alim [23] investigated the problem of developing laminar CHT of copper water nanofluid in a rectangular enclosure. Malvandi et al. [24] studied fluid flow and heat transfer of nanofluids over a flat plate with conjugate heat transfer by including the fluid effects of thermal resistance of the plate in the formulation. Zahan et al. [25] also studied the problem of MHD conjugate natural convection flow in a rectangular frame filled with a copper water nanofluid. Alsabery et al. [26] investigated the conjugate natural convection of $Al_2O_3$-water nanofluid in a square cavity using Buongiorno's two-phase model. Amongst their findings, they reported that, when the heat conduction is dominated, the heat transfer is increased with the increment of the nanoparticles volume fraction.

High temperatures are required to perform many engineering processes. Nuclear power plants, gas turbines, missiles, satellites, different types of equipment for aircraft, to name a few, can be included in such processes. Accordingly, radiation heat transfer knowledge is very important to design relevant devices. Furthermore, radiation has a significant effect on MHD flow and heat transfer characteristics from an industrial point of view. Industrial applications of thermal radiation include polymer technology, food production, engineering and spinning of fibers and advanced energy conversion in heat transfer at high temperatures. The effect of thermal radiation on MHD convection flow has been investigated by many researchers in the case of regular fluids and nanofluids. Takhar et al. [27] studied the effect of radiation on natural convection flow and heat transfer for a semi-infinite vertical plate with the transverse magnetic field. Emad [28] investigated free convection heat transfer characteristics of an electrically conducting fluid along an isothermal sheet with a transverse magnetic field. In this analysis, the simultaneous effects of buoyancy and radiation with internal heat generation or absorption were considered over the linearly stretched sheet taking into account a uniform free stream of constant velocity and temperature. El-Naby et al. [29] investigated natural convection unsteady flow over a semi-finite vertical plate with variable temperature, radiation, and transverse magnetic field. Ali et al. [30] investigated thermal radiation effects on the time-independent hydromagnetic forced convective flow of an electrically conducting and heat generating-absorbing fluid over a non-isothermal wedge. Mbeledogu et al. [31] obtained the perturbation solutions of the problem formed by the simultaneous action of buoyancy and transverse magnetic field on free convection flow of compressible Boussinesq fluid past a moving vertical plate. The viscosity and thermal conductivity of the fluid were a function of temperature and the radiative flux was confirmed using the Rosseland approximation. Ali et al. [32] analyzed the effect of thermal radiation and heat generation on viscous Joule heating MHD-conjugate heat transfer along a vertical flat plate. Their results showed that thermal radiation, viscous Joule heating and internal heat generation in the presence of conduction effects have a significant effect on MHD natural convection flow and thermal fields. In the case of nanofluids, Elazem et al. [33] considered the effect of radiation on the steady MHD flow and heat transfer of *Cu*–water and *Ag*–water nanofluids flow over a stretching sheet. Raju et al. [34] investigated the influence of the magnetic field, radiation, and non-uniform heat source/sink on Cu–Ethyline glycol and Ag–Ethyline glycol nanofluids flow over a moving vertical plate in a porous medium. The results from these studies reveal that, as thermal radiation increases, the rate of energy transported to the fluid increases, consequently an increase in temperature occurs.

The thermal radiation effect on the MHD-conjugate flow of nanofluids over flat plates with internal heat generation can be important in many industrial and theoretical applications. However, the literature review shows that no significant study investigated the combined effects of thermal radiation and heat generation on MHD-conjugate heat transfer flow on natural convection in a nanofluid filled enclosure. The objective of this study is to extend the work of Yu and Lin [9] by analyzing the conjugate heat transfer in MHD free convective flow of Cu–water and Ag–water

nanofluids along the vertical and horizontal plates with internal heat generation and thermal radiation. This study is theoretical and can have practical significance in designing and operation of plate heat exchangers. It is worth mentioning that the problem considered has applications in industries such as flat fins and cooling of electronic boards due to the inclusion of nanoparticles. The flow is subject to a uniform magnetic field imposed along the direction of the flow. We further demonstrate the application of efficient overlapping multi-domain bivariate spectral quasilinearisation method in solving a nonlinear system of partial differential equations (PDEs) modeling CHT problems. This method is more accurate than the non-overlapping multi-domain bivariate spectral quasilinearisation method (MD-BSQLM) [35]. The non-overlapping MD-BSQLM applies the multi-domain technique only in the time interval. However, the method considered in the present work applies the multi-domain technique in both space and time intervals. In addition to that, the method uses the overlapping multi-domain technique in the space interval. The overlapping grid strategy can improve the accuracy of spectral collocation based methods. The accuracy improvement is achieved through making the coefficient matrix in the matrix equation (resulting from the collocation process) less dense. This means that the coefficient matrix will be sparse. The sparsity of matrices caused by overlapping sub-domains can help to minimize the storage of large matrices and make it easy to perform matrix-vector multiplications. This is because there will be a lot of multiplication by zero which reduces the computational time and enables the matrices to be stored efficiently. Since the method combines the bivariate spectral quasilinearisation method [36], non-overlapping and overlapping multi-domain technique, for reference purposes, we shall refer to the method as the overlapping multi-domain bivariate spectral quasilinearisation method (OMD-BSQLM). The use of spectral collocation-based methods such as the OMD-BSQLM for solving systems of PDEs can be a most promising tool in the study of conjugate heat transfer problems. From the literature review, several studies [2,9] concluded that it is very difficult to obtain analytical solutions of conjugate heat transfer problems due to the matching conditions at the solid–fluid interface. These studies proposed the use of numerical methods such as finite difference schemes as the most promising procedures for performing this matching. However, the finite difference methods have a lot of limitations when compared to spectral methods. Spectral methods are highly accurate and more efficient than traditional methods such as the finite difference methods [37]. When applied to problems with smooth solutions, they use few grid points and require minimal computational time to generate accurate solutions, thus they are better than traditional methods. The spectral method algorithm is easy to implement in scientific computing software. To establish the accuracy of the OMD-BSQLM, certain limiting solutions of the flow equations are studied.

## 2. Mathematical Formulation

Let us consider the viscous, steady, incompressible, electrically conducting and free convection flow of nanofluid over a vertical flat plate and a horizontal flat plate of finite length $l$ and thickness $b$. The thickness of the plates is assumed to be smaller than the length. The base fluid is water and nanoparticles (Cu and Ag) are in thermal equilibrium with no slip between them. The thermophysical properties of the base fluid and different nanoparticles are shown in Table 1. It is assumed that the left side of the vertical plate and the lower side of the horizontal plate are maintained at the constant temperature $T_b$, such that $T_b > T_\infty$, where $T_\infty$ is the temperature of the ambient nanofluid. Heat is transferred by conduction from the outside surface of the solid plates coupled with the free convection in the nanofluid, while the axial heat conduction in the plates is neglected. A uniform magnetic field $B(x)$ is imposed along the direction of flow. The applied transverse magnetic field can be chosen in its special form as $B(x) = B_0 \alpha_f^{1/2} x^{-1}$, where $B_0$ is the steady strength of the magnetic field towards the $y$-axis. It is assumed that the induced magnetic field and the external electric field are negligible. Thermal radiation and internal heat generation terms are included in the energy equation. The geometry and coordinate system for the vertical and horizontal flat plates are shown in Figure 1.

With the above assumptions, equations of the conservation of mass, momentum, and energy are given by

$$\frac{\partial u}{\partial x} + \frac{\partial v}{\partial y} = 0, \tag{1}$$

$$u\frac{\partial u}{\partial x} + v\frac{\partial u}{\partial y} = -\frac{1}{\rho_{nf}}\frac{\partial p}{\partial x} + \nu_{nf}\frac{\partial^2 u}{\partial y^2} + g\beta_{nf}(T - T_\infty)\sin\varphi - \frac{\sigma_{nf}B^2(x)}{\rho_{nf}}u, \tag{2}$$

$$0 = -\frac{1}{\rho_{nf}}\frac{\partial p}{\partial y} + g\beta_{nf}(T - T_\infty)\cos\varphi, \tag{3}$$

$$u\frac{\partial T}{\partial x} + v\frac{\partial T}{\partial y} = \frac{k_{nf}}{(\rho C_p)_{nf}}\frac{\partial^2 T}{\partial y^2} + \frac{Q_0}{(\rho C_p)_{nf}}(T - T_\infty) - \frac{1}{(\rho C_p)_{nf}}\frac{\partial q_r}{\partial y}, \tag{4}$$

where $u$ and $v$ are the velocity components in the $x$- and $y$- directions, $p$ is the pressure, $g$ is the acceleration due to gravity, $T$ is the fluid temperature near the plate, $q_r$ is the radiative heat flux, $Q_0$ is the rate of heat generation, $\sigma_{nf}$ is the electrical conductivity, $\nu_{nf}$ is the kinematic viscosity, $\mu_{nf}$ is the dynamic viscosity, $\rho_{nf}$ is the effective density, $\alpha_{nf}$ is the thermal diffusivity, $k_{nf}$ is the effective thermal conductivity, $\beta_{nf}$ is the thermal expansion coefficient and $(\rho C_p)_{nf}$ is the heat capacity of the nanofluid. The term $Q_0(T - T_\infty)$ represents the amount of heat generated or absorbed per unit volume, where $Q_0$ is a constant which may be either positive for a heat sink or negative for a heat source. The radiative heat flux $q_r$ with Rosseland approximation has the form $q_r = -\frac{4\sigma^*}{3k^*}\frac{\partial T^4}{\partial y}$, where $\sigma^*$ is the Stefan–Boltzmann constant and $k^*$ is the mean absorption coefficient. The temperature differences within the flow are assumed to be sufficiently small such that $T^4$ may be expressed as a linear function of temperature. Expanding $T^4$ using Taylor series and neglecting higher order terms yields $T^4 \cong 4T_\infty^3 T - 3T_\infty^4$. The nanofluid constants are defined as [38–40]

$$\nu_{nf} = \frac{\mu_{nf}}{\rho_{nf}}, \; \mu_{nf} = \frac{\mu_f}{(1-\phi)^{2.5}}, \; \frac{k_{nf}}{k_f} = \left[\frac{(k_s + 2k_f) - 2\phi(k_f - k_s)}{(k_s + 2k_f) + \phi(k_f - k_s)}\right], \; \frac{\sigma_{nf}}{\sigma_f} = \left[1 + \frac{3\left(\frac{\sigma_s}{\sigma_f} - 1\right)\phi}{\left(\frac{\sigma_s}{\sigma_f} + 2\right) - \left(\frac{\sigma_s}{\sigma_f} - 1\right)\phi}\right],$$

$$\rho_{nf} = (1-\phi)\rho_f + \phi\rho_s, \; (\rho C_p)_{nf} = (1-\phi)(\rho C_p)_f + \phi(\rho C_p)_s, \; \beta_{nf} = (1-\phi)\beta_f + \phi\beta_s, \; \alpha_{nf} = \frac{k_{nf}}{(\rho C_p)_{nf}}, \tag{5}$$

where $\phi$ is the solid volume fraction of nanoparticles, $\beta$ is the thermal expansion, subscripts $_{f,s}$ and $_{nf}$ denote fluid, solid and nanofluid, respectively. For the vertical plate, the angle $\varphi$ is $\pi/2$ and $\partial p/\partial x$, and $\partial p/\partial y$ are both equal to zero. For the horizontal plate, the angle $\varphi$ to the horizontal is equal to zero. In formulating Equations (1)–(4), viscous dissipation and compression work have been neglected. Moreover, the physical properties of the fluid are assumed to be constant except for the density variation that induces a buoyancy force. The boundary conditions for Equations (1)–(4) are

$$u = 0, \; v = 0, \quad \text{at} \quad y = 0, \tag{6}$$

$$u \to 0, \; p \to 0, \quad T \to T_\infty \quad \text{as} \quad y \to \infty. \tag{7}$$

**Table 1.** Thermophysical properties of the base fluid and the nanoparticles [38].

| | Base Fluid | Nanoparticles | |
| --- | --- | --- | --- |
| Physical Properties | Water | Copper (Cu) | Silver (Ag) |
| $C_p$ (J/kgK) | 4179 | 385 | 235 |
| $\rho$ (Kg/m$^3$) | 997.1 | 8933 | 10,500 |
| k (W/mK) | 0.613 | 401 | 429 |
| $\sigma$ (Sm$^{-1}$) | 0.05 | $5.96 \times 10^7$ | $6.3 \times 10^7$ |
| $\beta \times 10^5$ (K$^{-1}$) | 21 | 1.67 | 1.89 |

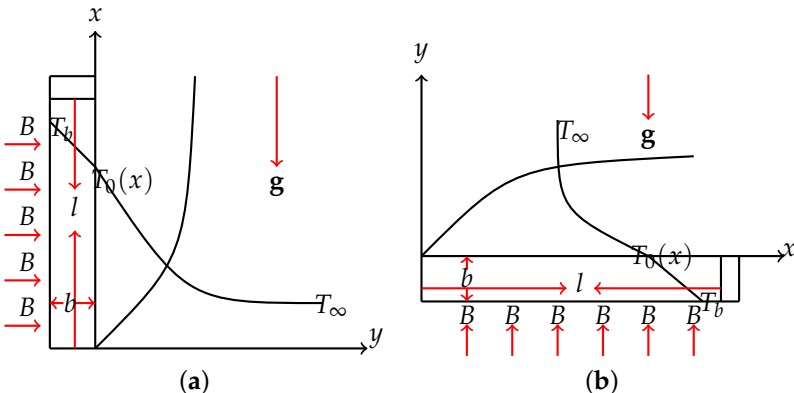

**Figure 1.** Physical model and coordinate system. (**a**) vertical plate; (**b**) horizontal plate.

## 2.1. Dimensionless Equations for the Vertical Plate

Yu and Lin [9] developed the following non-dimensional variables:

$$\psi(x,y) = \alpha_f \lambda f(\xi,\eta), \quad \eta(x,y) = (y/x)\lambda, \; \xi(x) = \left[1 + \sigma R_{a_t}/(\sigma R_{a_h})^{4/5}\right]^{-1}, \quad \theta(\xi,\eta) = \frac{T - T_\infty}{T_b - T_\infty}\xi^{-1}, \; (8)$$

where $R_{a_t} = g\beta(T_b - T_\infty)x^3/\alpha_f \nu$ is the Rayleigh number, $R_{a_h} = g\beta(q_h x/k_r)x^3/\alpha_f \nu$ is the Reyleigh number for a plate with constant wall flux $q_h = k_s(T_b - T_\infty)/b$, $\lambda = \left[(\sigma R_{a_t})^{-1} + (\sigma R_{a_h})^{-4/5}\right]^{-1/4}$, $\sigma = Pr/(1 + Pr)$, $\psi(x,y)$ is the stream function defined by $u = \partial\psi/\partial y$ and $v = -\partial\psi/\partial x$, $\eta(x,y)$ and $\xi(x)$ are the dimensionless coordinates, $f(\xi,\eta)$ is the dimensionless stream function and $\theta(\xi,\eta)$ is the dimensionless temperature. By using Equation (8), Equations (1)–(4) along with the boundary conditions (6) and (7) are reduced to the following two-point boundary value problem:

$$Prf''' + \phi_1\left[\frac{16-\xi}{20}ff'' - \frac{6-\xi}{10}f'^2 - M^2\phi_2 f' + \phi_3(1+Pr)\theta\right] = \frac{\phi_1}{5}\xi(1-\xi)\left[f'\frac{\partial f'}{\partial\xi} - f''\frac{\partial f}{\partial\xi}\right], \quad (9)$$

$$\left(1 + \frac{k_f}{k_{nf}}Rd\right)\theta'' + \phi_4\left[\frac{16-\xi}{20}f\theta' - \frac{1-\xi}{5}f'\theta\right] + \frac{k_f}{k_{nf}}Q\xi\theta = \frac{\phi_4}{5}\xi(1-\xi)\left[f'\frac{\partial\theta}{\partial\xi} - \theta'\frac{\partial f}{\partial\xi}\right], \quad (10)$$

where $M^2 = \frac{\sigma_f B_0^2}{\rho_f \lambda^2}$ is the magnetic field parameter, $Pr = \frac{\nu_f}{\alpha_f}$ is the Prandtl number, $Rd = \frac{16\sigma^* T_\infty^3}{3k^* k_f}$ is the radiation parameter and $Q = \frac{Q_0 x^2}{k_f \lambda^2}$ is the heat generation parameter. The nanoparticle volume fractions $\phi_1, \phi_2, \phi_3$ and $\phi_4$ depend on the thermal properties of the nanofluid and are defined as

$$\phi_1 = [1-\phi]^{2.5}\left(1 - \phi + \phi\frac{\rho_s}{\rho_f}\right), \; \phi_2 = \left(1 + \frac{3(\sigma_s/\sigma_f - 1)\phi}{(\sigma_s/\sigma_f + 2) - (\sigma_s/\sigma_f - 1)\phi}\right)\frac{1}{\left((1-\phi) + \phi(\frac{\rho_s}{\rho_f})\right)},$$

$$\phi_3 = (1-\phi) + \phi(\beta_s/\beta_f), \quad \phi_4 = \left[\frac{k_s + 2k_f + \phi(k_f - k_s)}{k_s + 2k_f - 2\phi(k_f - k_s)}\right]\left((1-\phi) + \phi\frac{(\rho C_p)_s}{(\rho C_p)_f}\right). \quad (11)$$

The corresponding boundary conditions in dimensionless form are

$$f(\xi,0) = 0, \; f'(\xi,0) = 0, \; \xi\theta(\xi,0) - (1-\xi)^{5/4}\theta'(\xi,0) = 1,$$
$$f'(\xi,\infty) = 0, \; \theta(\xi,\infty) = 0. \quad (12)$$

### 2.2. Dimensionless Equations for the Horizontal Plate

Using the following non-dimensional variables in [9], namely

$$\psi(x,y) = \alpha_f \lambda f(\xi,\eta), \quad \eta(x,y) = (y/x)\lambda, \quad \xi(x) = \left[1 + \sigma Ra_t/(\sigma Ra_h)^{5/6}\right]^{-1.5}, \tag{13}$$

$$\theta(\xi,\eta) = \frac{T - T_\infty}{T_b - T_\infty}\xi^{-1}, \quad \omega(\xi,\eta) = \sigma p x^2/\rho\alpha_f \nu\lambda^4, \tag{14}$$

where $\lambda = \left[(\sigma Ra_t)^{-1} + (\sigma Ra_h)^{-5/6}\right]^{-1.5}$ and $\omega(\xi,\eta)$ is the dimensionless pressure. Equations (1)–(4) along with their boundary conditions (6) and (7) for the horizontal plate are reduced to

$$Pr f''' + \phi_1 \left[\frac{10-\xi}{15}ff'' - \frac{5-2\xi}{15}f'^2 - \phi_2 M^2 f'\right] + \frac{(1-\phi)^{2.5}}{15}(1+Pr)[(5+\xi)\eta\omega' - (10-4\xi)\omega]$$
$$= \frac{\phi_1}{3}\xi(1-\xi)\left[f'\frac{\partial f'}{\partial \xi} - f''\frac{\partial f}{\partial \xi} + (1+Pr)\frac{\partial \omega}{\partial \xi}\right], \tag{15}$$

$$\omega' = \theta, \tag{16}$$

$$\left(1 + \frac{k_f}{k_{nf}}Rd\right)\theta'' + \phi_4\left[\frac{10-\xi}{15}f\theta' - \frac{1-\xi}{3}f'\theta\right] + \frac{k_f}{k_{nf}}Q\xi\theta = \frac{\phi_4}{3}\xi(1-\xi)\left[f'\frac{\partial \theta}{\partial \xi} - \theta'\frac{\partial f}{\partial \xi}\right]. \tag{17}$$

The corresponding boundary conditions in dimensionless form are

$$f(\xi,0) = 0, \quad f'(\xi,0) = 0, \quad \xi\theta(\xi,0) - (1-\xi)^{6/5}\theta'(\xi,0) = 1, \tag{18}$$

$$f'(\xi,\infty) = 0, \quad \theta(\xi,\infty) = 0, \quad \omega(\xi,\infty) = 0. \tag{19}$$

## 3. Solution Procedure

In this section, we describe the application of the OMD-BSQLM to find numerical solutions of the transformed nonlinear PDEs. The method uses the overlapping multi-domain technique, Chebyshev–Gauss–Lobatto grid points [41,42], and the quasilinearisation method [43], together with spectral collocation on approximate functions defined as bivariate Lagrange interpolation polynomials. The multi-domain approach divides the time interval into non-overlapping sub-intervals and the space interval into overlapping sub-intervals. The quasilinearisation technique helps to linearise the nonlinear PDEs. The spectral collocation method is applied independently both in space and time variables in the linearized equations. In order to apply the OMD-BSQLM, the time interval $\xi \in [0, \xi_F]$ is decomposed into $q$ non-overlapping sub-intervals defined as

$$J_v = (\xi_{v-1}, \xi_v), \quad v = 1, 2, 3, \ldots, q, \quad \text{with} \quad 0 = \xi_0 < \xi_1 < \xi_2 < \cdots < \xi_{q-1} < \xi_q = \xi_F, \tag{20}$$

For the semi-finite space domain $[0, \infty)$, a truncated grid $[0, \eta_\infty]$ is used. We choose a finite value of $\eta_\infty$ that is large enough such that the flow properties at $\eta_\infty$ resemble those at $\infty$. The truncated space interval $[0, \eta_\infty]$ is decomposed into $p$ overlapping sub-intervals of length $L$, denoted by

$$I_\mu = [\eta_0^\mu, \eta_{N_\eta}^\mu], \quad \mu = 1, 2, 3, \ldots, p, \tag{21}$$

where each $I_\mu$ interval is further discretized into $N_\eta + 1$ collocation points. Without loss of generality, we will consider that each subinterval has the same length given by

$$L = \frac{\eta_\infty}{p + \frac{1}{2}(1-p)(1 - \cos\frac{\pi}{N_\eta})} \tag{22}$$

for the overlap to be possible, and the same number of collocation points $(N_\eta + 1)$ is used in each subinterval. In the domain decomposition scheme, we use overlapping subintervals $I_\mu$, where the

first two points of the interval $I_{\mu+1}$ coincide with the last two points of the interval $I_\mu$, that is, $\eta_0^1 = 0$, $\eta_{N_\eta}^p = \eta_\infty, \eta_{N_\eta-1}^\mu = \eta_0^{\mu+1}$ and $\eta_{N_\eta}^\mu = \eta_1^{\mu+1}$. The non-overlapping and overlapping multi-domain grids are shown in Figures 2 and 3, respectively.

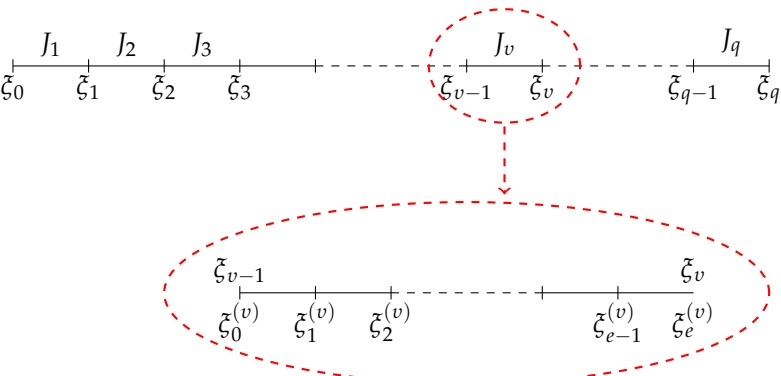

**Figure 2.** Non-overlapping grid ($\xi$-domain).

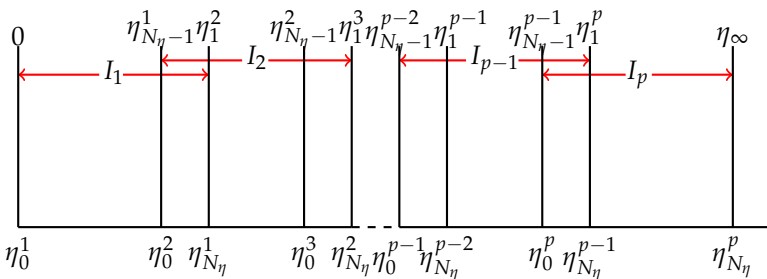

**Figure 3.** Overlapping grid ($\eta$-domain).

### 3.1. Numerical Solution for the Vertical Plate

Applying the quasilinearisation method in each sub-interval to Equations (9) and (10) gives the following system of linear PDEs:

$$\alpha_{1,3,r}^{(1,\mu,v)} \frac{\partial^3 f_{r+1}^{(\mu,v)}}{\partial \eta^3} + \alpha_{1,2,r}^{(1,\mu,v)} \frac{\partial^2 f_{r+1}^{(\mu,v)}}{\partial \eta^2} + \alpha_{1,1,r}^{(1,\mu,v)} \frac{\partial f_{r+1}^{(\mu,v)}}{\partial \eta} + \alpha_{1,0,r}^{(1,\mu,v)} f_{r+1}^{(\mu,v)} + \alpha_{2,0,r}^{(1,\mu,v)} \theta_{r+1}^{(\mu,v)}$$

$$+ \gamma_{1,r}^{(1,\mu,v)} \frac{\partial}{\partial \xi} \left( \frac{\partial f_{r+1}^{(\mu,v)}}{\partial \eta} \right) + \beta_{1,r}^{(1,\mu,v)} \frac{\partial f_{r+1}^{(\mu,v)}}{\partial \xi} = R_{1,r}^{(\mu,v)}, \tag{23}$$

$$\alpha_{2,2,r}^{(2,\mu,v)} \frac{\partial^2 \theta_{r+1}^{(\mu,v)}}{\partial \eta^2} + \alpha_{2,1,r}^{(2,\mu,v)} \frac{\partial \theta_{r+1}^{(\mu,v)}}{\partial \eta} + \alpha_{2,0,r}^{(2,\mu,v)} \theta_{r+1}^{(\mu,v)} + \alpha_{1,1,r}^{(2,\mu,v)} \frac{\partial f_{r+1}^{(\mu,v)}}{\partial \eta} + \alpha_{1,0,r}^{(2,\mu,v)} f_{r+1}^{(\mu,v)}$$

$$+ \beta_{2,r}^{(2,\mu,v)} \frac{\partial \theta_{r+1}^{(\mu,v)}}{\partial \xi} + \beta_{1,r}^{(2,\mu,v)} \frac{\partial f_{r+1}^{(\mu,v)}}{\partial \xi} = R_{2,r}^{(\mu,v)}, \tag{24}$$

where the variable coefficients are given by

$$\alpha_{1,3,r}^{(1,\mu,v)} = Pr, \quad \alpha_{1,2,r}^{(1,\mu,v)} = \frac{\phi_1(16-\xi)}{20} f_r^{(\mu,v)} + \frac{\phi_1\xi(1-\xi)}{5} \frac{\partial f_r^{(\mu,v)}}{\partial\xi}, \quad \alpha_{1,0,r}^{(1,\mu,v)} = \frac{\phi_1(16-\xi)}{20} \frac{\partial^2 f_r^{(\mu,v)}}{\partial\eta^2},$$

$$\alpha_{1,1,r}^{(1,\mu,v)} = -\frac{\phi_1(6-\xi)}{5} \frac{\partial f_r^{(\mu,v)}}{\partial\eta} - M\phi_1\phi_2 - \frac{\phi_1\xi(1-\xi)}{5} \frac{\partial}{\partial\xi}\left(\frac{\partial f_r^{(\mu,v)}}{\partial\eta}\right), \quad \alpha_{2,0,r}^{(1,\mu,v)} = \phi_1\phi_3(1+Pr),$$

$$\alpha_{2,2,r}^{(2,\mu,v)} = 1 + \left(1 + \frac{k_f}{k_{nf}}Rd\right), \quad \alpha_{2,0,r}^{(2,\mu,v)} = \frac{k_k}{k_{nf}}Q\xi - \frac{\phi_4(1-\xi)}{5} \frac{\partial f_r^{(\mu,v)}}{\partial\eta}, \quad \alpha_{2,1,r}^{(2,\mu,v)} = \frac{\phi_4(16-\xi)}{20} f_r^{(\mu,v)}$$

$$+\frac{\phi_4\xi(1-\xi)}{5} \frac{\partial f_r^{(\mu,v)}}{\partial\xi}, \quad \alpha_{1,1,r}^{(2,\mu,v)} = -\frac{\phi_4(1-\xi)}{5} \theta_r^{(\mu,v)} - \frac{\phi_4\xi(1-\xi)}{5} \frac{\partial\theta_r^{(\mu,v)}}{\partial\xi}, \quad \alpha_{1,0,r}^{(2,\mu,v)} = \frac{\phi_4(16-\xi)}{20} \frac{\partial\theta_r^{(\mu,v)}}{\partial\eta},$$

$$\gamma_{1,r}^{(1,\mu,v)} = -\frac{\phi_1\xi(1-\xi)}{5} \frac{\partial f_r^{(\mu,v)}}{\partial\eta}, \quad \beta_{1,r}^{(1,\mu,v)} = \frac{\phi_1\xi(1-\xi)}{5} \frac{\partial^2 f_r^{(\mu,v)}}{\partial\eta^2}, \quad \beta_{2,r}^{(2,\mu,v)} = -\frac{\phi_4\xi(1-\xi)}{5} \frac{\partial f_r^{(\mu,v)}}{\partial\eta},$$

$$\beta_{1,r}^{(2,\mu,v)} = \frac{\phi_4\xi(1-\xi)}{5} \frac{\partial\theta_r^{(\mu,v)}}{\partial\eta}, \quad R_{1,r}^{(\mu,v)} = \frac{\phi_1(16-\xi)}{20} f_r^{(\mu,v)} \frac{\partial^2 f_r^{(\mu,v)}}{\partial\eta^2} - \frac{\phi_1(6-\xi)}{10} \left(\frac{\partial f_r^{(\mu,v)}}{\partial\eta}\right)^2$$

$$-\frac{\phi_1\xi(1-\xi)}{5} \frac{\partial f_r^{(\mu,v)}}{\partial\eta} \frac{\partial}{\partial\xi}\left(\frac{\partial f_r^{(\mu,v)}}{\partial\eta}\right) + \frac{\phi_1\xi(1-\xi)}{5} \frac{\partial^2 f_r^{(\mu,v)}}{\partial\eta^2} \frac{\partial f_r^{(\mu,v)}}{\partial\xi}, \quad R_{2,r}^{(\mu,v)} = \frac{\phi_4(16-\xi)}{20} f_r^{(\mu,v)} \frac{\partial\theta_r^{(\mu,v)}}{\partial\eta}$$

$$-\frac{\phi_4(1-\xi)}{5} \theta_r^{(\mu,v)} \frac{\partial f_r^{(\mu,v)}}{\partial\eta} - \frac{\phi_4\xi(1-\xi)}{5} \frac{\partial f_r^{(\mu,v)}}{\partial\eta} \frac{\partial\theta_r^{(\mu,v)}}{\partial\xi} + \frac{\phi_4\xi(1-\xi)}{5} \frac{\partial\theta_r^{(\mu,v)}}{\partial\eta} \frac{\partial f_r^{(\mu,v)}}{\partial\xi},$$

subject to boundary conditions

$$f_{r+1}^{(\mu,v)}(\xi,0) = 0, \quad \frac{\partial f_{r+1}^{(\mu,v)}}{\partial\eta}(\xi,0) = 0, \quad \xi\theta_{r+1}^{(\mu,v)}(\xi,0) - (1-\xi)^{5/4}\frac{\partial\theta_{r+1}^{(\mu,v)}}{\partial\eta}(\xi,0) = 1,$$

$$\frac{\partial f_{r+1}^{(\mu,v)}}{\partial\eta}(\xi,\infty) = 0, \quad \theta_{r+1}^{(\mu,v)}(\xi,\infty) = 0. \tag{25}$$

The constants $r$ and $r + 1$ denote previous and current iterations, respectively. The system of linear PDEs (23) and (24) is discretized using the spectral collocation method in both $\eta$ and $\xi$ directions. Before applying the spectral method on the sub-intervals, the time interval $\xi \in [\xi_{v-1}, \xi_v]$ is transformed to $\tau \in [-1, 1]$ using the linear transformation

$$\xi_j^v = \frac{1}{2}(\xi_v - \xi_{v-1})\tau_j + \frac{1}{2}(\xi_v + \xi_{v-1}), \quad \tau_j = \cos\left(\frac{\pi j}{N_\xi}\right), \tag{26}$$

and the space region $\eta \in [\eta_0^\mu, \eta_{N_\eta}^\mu]$ is transformed to $z \in [-1, 1]$ using the linear transformation

$$\eta_i^\mu = \frac{L}{2}(z_i + 1), \quad z_i = \cos\left(\frac{\pi i}{N_\eta}\right). \tag{27}$$

We assume that, at each sub-interval, the required solution, say $f(\eta, \xi)$, can be approximated by a bivariate Lagrange interpolation polynomial of the form

$$f^{(\mu,v)}(\eta,\xi) \approx \sum_{i=0}^{N_\eta}\sum_{j=0}^{N_\xi} f^{(\mu,v)}(z_i,\tau_j)\mathcal{L}_i(z)\mathcal{L}_j(\tau), \tag{28}$$

for $\mu = 1, 2, 3, \ldots, p$ and $v = 1, 2, 3, \ldots, q$. The bivariate interpolation polynomial interpolates $f^{(\mu,v)}(z, \tau)$ at selected points $(z_i, \tau_j)$ in both $z$ and $\tau$ directions, for $i = 0, 1, 2, \ldots, N_\eta$ and $j = 0, 1, 2, \ldots, N_\xi$. The function $\mathcal{L}_i(z)$ and $\mathcal{L}_j(\tau)$ are the well known characteristic Lagrange cardinal polynomial based on the Chebyshev–Gauss–Lobatto points. The required solution for $\theta(\eta, \xi)$ can be approximated in a similar manner. The solution procedure requires that the derivatives of $\mathcal{L}_i(z)$ and

$\mathcal{L}_j(\tau)$ with respect to $z$ and $\tau$, respectively be evaluated at the Chebyshev–Gauss–Lobatto grid points. The derivatives of $f^{(\mu,v)}(\eta,\xi)$ with respect to $\eta$ and $\xi$ at the Chebyshev–Gauss–Lobatto points $(z_k,\tau_i)$, are computed as

$$\left.\frac{\partial f^{(\mu,v)}}{\partial \eta}\right|_{(z_k,\tau_i)} = \sum_{\omega}^{N_\eta}\sum_{j=0}^{N_\xi} f^{(\mu,v)}(z_\omega,\tau_j)\frac{d\mathcal{L}_\omega(z_k)}{dz}L_j(\tau_i) \tag{29}$$

$$= \sum_{\omega=0}^{N_\eta} D_{k,\omega}^{(\mu)} f^{(\mu,v)}(z_\omega,\tau_i) = \left[\mathbf{D}^{(\mu)}\right]\mathbf{F}_i^{(\mu,v)}, \tag{30}$$

$$\left.\frac{\partial f^{(\mu,v)}}{\partial \xi}\right|_{(z_k,\tau_i)} = \sum_{\omega}^{N_\eta}\sum_{j=0}^{N_\xi} f^{(\mu,v)}(z_\omega,\tau_j)\mathcal{L}_\omega(z_k)\frac{d\mathcal{L}_j(\tau_i)}{d\tau}$$

$$= \sum_{j=0}^{N_\xi} d_{i,j} f^{(\mu,v)}(z_k,\tau_j) = \sum_{j=0}^{N_\xi} d_{i,j}\mathbf{F}_j^{(\mu,v)}, \tag{31}$$

where $d_{i,j} = \frac{d\mathcal{L}_j(\tau_i)}{d\tau}$ is the $i$th and $j$th entry of the standard first derivative Chebyshev–Gauss–Lobatto based differentiation matrix $\mathbf{d} = [d_{i,j}]$, for $i,j = 0,1,2,3,\ldots,N_\xi$, of size $(N_\xi + 1) \times (N_\xi + 1)$, $D_{k,\omega}^{(\mu)} = \frac{2}{\eta_{N_\eta}^\mu - \eta_0^\mu}D_{k,\omega}$ with $D_{k,\omega} = \frac{d\mathcal{L}_\omega(z_k)}{dz}$ being the $k$th and $\omega$th entries of the standard first derivative Chebyshev–Gauss–Lobatto differentiation matrix of size $(M+1) \times (M+1)$, where $M = N_\eta + (N_\eta - 1)(p-1)$ is the total number of collocation points over a single domain $[-1,1]$. In general, to find an $s$th order derivative with respect to $\eta$, we have

$$\left.\frac{\partial^s f^{(\mu,v)}}{\partial \eta^s}\right|_{(z_k,\tau_i)} = \sum_{\omega=0}^{N_\eta}\left[D_{k,\omega}^{(\mu)}\right]^s f^{(\mu,v)}(z_\omega,\tau_i) = \left[\mathbf{D}^{(\mu)}\right]^s\mathbf{F}_i^{(\mu,v)}. \tag{32}$$

The vector $\mathbf{F}_i^{(\mu,v)}$ is defined as

$$\mathbf{F}_i^{(\mu,v)} = \left[f^{(\mu,v)}(z_0^{(\mu)},\tau_i^{(v)}), f^{(\mu,v)}(z_1^{(\mu)},\tau_i^{(v)}), f^{(\mu,v)}(z_2^{(\mu)},\tau_i^{(v)}),\ldots, f^{(\mu,v)}(z_{N_\eta}^{(\mu)},\tau_i^{(v)})\right]^T, \tag{33}$$

where $T$ denotes the matrix transpose. The derivatives $d$ and $\left[\mathbf{D}^{(\mu)}\right]^s$ are scaled by multiplying by the factors $\Lambda = \frac{2}{\xi_v - \xi_{v-1}}$ and $\Omega^s = \left(\frac{2}{\eta_{N_\eta}^\mu - \eta_0^\mu}\right)^s = \left(\frac{2}{L}\right)^s$, respectively. The space and time derivatives of $\theta$ at each sub-interval can be transformed to discrete matrix form in a similar manner. Applying the spectral collocation method by evaluating Equations (23) and (24) at the collocation points and making use of the derivative matrices as well as incorporating the initial condition which corresponds to $\xi_{N_\xi} = -1$ gives

$$A_{1,1}^{(\mu,v)}\mathbf{F}_{i,r+1}^{(\mu,v)} + A_{1,2}^{(\mu,v)}\mathbf{\Theta}_{i,r+1}^{(\mu,v)} + \gamma_{1,r}^{(1,\mu,v)}\sum_{j=0}^{N_\xi-1} d_{i,j}\mathbf{D}^{(\mu)}\mathbf{F}_j^{(\mu,v)} + \beta_{1,r}^{(1,\mu,v)}\sum_{j=0}^{N_\xi-1} d_{i,j}\mathbf{F}_j^{(\mu,v)} = \mathbf{K}_{1,i}^{(\mu,v)}, \tag{34}$$

$$A_{2,1}^{(\mu,v)}\mathbf{F}_{i,r+1}^{(\mu,v)} + A_{2,2}^{(\mu,v)}\mathbf{\Theta}_{i,r+1}^{(\mu,v)} + \beta_{1,r}^{(2,\mu,v)}\sum_{j=0}^{N_\xi-1} d_{i,j}\mathbf{F}_j^{(\mu,v)} + \beta_{2,r}^{(2,\mu,v)}\sum_{j=0}^{N_\xi-1} d_{i,j}\mathbf{\Theta}_j^{(\mu,v)} = \mathbf{K}_{2,i}^{(\mu,v)}, \tag{35}$$

where

$$\mathbf{K}_{1,i}^{(\mu,v)} = \mathbf{R}_{1,i}^{(\mu,v)} - \gamma_{1,r}^{(1,\mu,v)} d_{i,N_\zeta}\mathbf{D}^{(\mu)}\mathbf{F}_{N_\xi}^{(\mu,v)} - \beta_{1,r}^{(1,\mu,v)} d_{i,N_\zeta}\mathbf{F}_{N_\xi}^{(\mu,v)},$$

$$\mathbf{K}_{2,i}^{(\mu,v)} = \mathbf{R}_{2,i}^{(\mu,v)} - \beta_{1,r}^{(2,\mu,v)} d_{i,N_\zeta}\mathbf{F}_{N_\xi}^{(\mu,v)} - \beta_{2,r}^{(2,\mu,v)} d_{i,N_\zeta}\mathbf{\Theta}_{N_\xi}^{(\mu,v)}$$

For $i = 0, 1, 2, \ldots N_{\xi}$, Equations (34) and (35) form an $N_{\xi}(M+1) \times N_{\xi}(M+1)$ matrix system

$$
\begin{bmatrix}
A_{0,0}^{(1,1,p,v)} & \cdots & A_{0,N_\xi}^{(1,1,p,v)} & & & & A_{0,0}^{(1,2,p,v)} & \cdots & A_{0,N_\xi}^{(1,2,p,v)} & & & \\
A_{1,0}^{(1,1,p,v)} & \cdots & A_{1,N_\xi}^{(1,1,p,v)} & & & & A_{1,0}^{(1,2,p,v)} & \cdots & A_{1,N_\xi}^{(1,2,p,v)} & & & \\
\ddots & \ddots & \ddots & & & & \ddots & \ddots & \ddots & & & \\
A_{N_\xi-1,0}^{(1,1,p,v)} & \cdots & A_{N_\xi-1,N_\xi}^{(1,1,p,v)} & & & & A_{N_\xi-1,0}^{(1,2,p,v)} & \cdots & A_{N_\xi-1,N_\xi}^{(1,2,p,v)} & & & \\
\end{bmatrix}
\begin{bmatrix}
\mathbf{F}_{0,r+1}^{(p,v)} \\ \mathbf{F}_{1,r+1}^{(p,v)} \\ \vdots \\ \mathbf{F}_{N_\xi-1,r+1}^{(p,v)} \\ \vdots
\end{bmatrix}
=
\begin{bmatrix}
\mathbf{K}_{1,0}^{(p,v)} \\ \mathbf{K}_{1,1}^{(p,v)} \\ \vdots \\ \mathbf{K}_{1,N_\xi-1}^{(p,v)} \\ \vdots
\end{bmatrix}, \qquad (36)
$$

Left matrix blocks (upper): entries $A_{j,k}^{(1,1,p,v)}$, $A_{j,k}^{(1,2,p,v)}$, $A_{j,k}^{(1,1,p-1,v)}$, $A_{j,k}^{(1,2,p-1,v)}$, $A_{j,k}^{(1,1,1,v)}$, $A_{j,k}^{(1,2,1,v)}$; (lower): $A_{j,k}^{(2,1,p,v)}$, $A_{j,k}^{(2,2,p,v)}$, $A_{j,k}^{(2,1,p-1,v)}$, $A_{j,k}^{(2,2,p-1,v)}$, $A_{j,k}^{(2,1,1,v)}$, $A_{j,k}^{(2,2,1,v)}$.

Solution vector: $\mathbf{F}_{i,r+1}^{(p,v)}, \ldots, \mathbf{F}_{i,r+1}^{(1,v)}, \Theta_{i,r+1}^{(p,v)}, \ldots, \Theta_{i,r+1}^{(1,v)}$.

Right-hand side vector: $\mathbf{K}_{1,k}^{(p,v)}, \ldots, \mathbf{K}_{1,k}^{(1,v)}, \mathbf{K}_{2,k}^{(p,v)}, \ldots, \mathbf{K}_{2,k}^{(1,v)}$.

where

$$A_{i,i}^{(1,1,p,v)} = \alpha_{1,3,r}^{(1,\mu,v)} \left[\mathbf{D}^{(\mu)}\right]^3 + \alpha_{1,2,r}^{(1,\mu,v)} \left[\mathbf{D}^{(\mu)}\right]^2 + \alpha_{1,1,r}^{(1,\mu,v)}\mathbf{D}^{(\mu)} + \alpha_{1,0,r}^{(1,\mu,v)} + \beta_{1,r}^{(1,\mu,v)}d_{i,i}\mathbf{I} + \gamma_{1,r}^{(1,\mu,v)}d_{i,i}\mathbf{D}^{(\mu)},$$

$$A_{i,i}^{(1,2,p,v)} = \alpha_{2,0,r}^{(1,\mu,v)}\mathbf{I}, \; A_{i,i}^{(2,1,p,v)} = \alpha_{1,1,r}^{(2,\mu,v)}\mathbf{D}^{(\mu)} + \alpha_{1,0,r}^{(2,\mu,v)} + \beta_{1,r}^{(2,\mu,v)}d_{i,i}\mathbf{I}, \tag{37}$$

$$A_{i,i}^{(2,2,p,v)} = \alpha_{2,2,r}^{(2,\mu,v)} \left[\mathbf{D}^{(\mu)}\right]^2 + \alpha_{2,1,r}^{(2,\mu,v)}\mathbf{D}^{(\mu)} + \alpha_{2,0,r}^{(2,\mu,v)} + \beta_{2,r}^{(2,\mu,v)}d_{i,i}\mathbf{I}, \quad \text{when } i = j$$

and

$$A_{i,j}^{(1,1,p,v)} = \beta_{1,r}^{(1,\mu,v)}d_{i,j}\mathbf{I} + \gamma_{1,r}^{(1,\mu,v)}d_{i,j}\mathbf{D}^{(\mu)}, \quad A_{i,j}^{(1,2,p,v)} = \mathbf{0}, \; A_{i,j}^{(2,1,p,v)} = \beta_{1,r}^{(2,\mu,v)}d_{i,j}\mathbf{I},$$

$$A_{i,j}^{(2,2,p,v)} = \beta_{2,r}^{(2,\mu,v)}d_{i,j}\mathbf{I}, \quad \text{when } i \neq j. \tag{38}$$

The vectors $\mathbf{F}_{i,r+1}^{(\mu,v)}$, and $\mathbf{\Theta}_{i,r+1}^{(\mu,v)}$ denote the values of $f$ and $\theta$ approximated at the collocation points, and $\mathbf{I}$ is the standard $(M+1) \times (M+1)$ identity matrix. Starting from suitable initial guesses, the numerical solution for $f(\eta, \xi)$ and $\theta(\eta, \xi)$ are obtained by solving matrix Equation (36) iteratively for $r = 1, 2, \ldots, \sigma$, where $\sigma$ is the number of iterations to be used.

### 3.2. Numerical Solution for the Horizontal Plate

Applying a quasilinearisation method in each subinterval to the system of nonlinear PDEs (15)–(17) gives the following system of linear PDEs:

$$\alpha_{1,3,r}^{(1,\mu,v)}\frac{\partial^3 f_{r+1}^{(\mu,v)}}{\partial \eta^3} + \alpha_{1,2,r}^{(1,\mu,v)}\frac{\partial^2 f_{r+1}^{(\mu,v)}}{\partial \eta^2} + \alpha_{1,1,r}^{(1,\mu,v)}\frac{\partial f_{r+1}^{(\mu,v)}}{\partial \eta} + \alpha_{1,0,r}^{(1,\mu,v)} f_{r+1}^{(\mu,v)} + \alpha_{2,1,r}^{(1,\mu,v)}\frac{\partial \omega_{r+1}^{(\mu,v)}}{\partial \eta},$$

$$+\alpha_{2,0,r}^{(1,\mu,v)}\omega_{r+1}^{(\mu,v)} + \gamma_{1,1,r}^{(1,\mu,v)}\frac{\partial}{\partial \xi}\left(\frac{\partial f_{r+1}^{(\mu,v)}}{\partial \eta}\right) + \beta_{1,1,r}^{(1,\mu,v)}\frac{\partial f_{r+1}^{(\mu,v)}}{\partial \xi} + \beta_{2,1,r}^{(1,\mu,v)}\frac{\partial \omega_{r+1}^{(\mu,v)}}{\partial \xi} = R_{1,r}^{(\mu,v)}, \tag{39}$$

$$\alpha_{2,1,r}^{(2,\mu,v)}\frac{\partial \omega_{r+1}^{(\mu,v)}}{\partial \eta} + \alpha_{3,0,r}^{(2,\mu,v)}\theta_{r+1}^{(\mu,v)} = 0, \tag{40}$$

$$\alpha_{3,2,r}^{(3,\mu,v)}\frac{\partial^2 \theta_{r+1}^{(\mu,v)}}{\partial \eta^2} + \alpha_{3,1,r}^{(3,\mu,v)}\frac{\partial \theta_{r+1}^{(\mu,v)}}{\partial \eta} + \alpha_{3,0,r}^{(3,\mu,v)}\theta_{r+1}^{(\mu,v)} + \alpha_{1,1,r}^{(3,\mu,v)}\frac{\partial f_{r+1}^{(\mu,v)}}{\partial \eta} + \alpha_{1,0,r}^{(3,\mu,v)} f_{r+1}^{(\mu,v)}$$

$$+\beta_{3,1,r}^{(3,\mu,v)}\frac{\partial \theta_{r+1}^{(\mu,v)}}{\partial \xi} + \beta_{1,1,r}^{(3,\mu,v)}\frac{\partial f_{r+1}^{(\mu,v)}}{\partial \xi} = R_{3,r}^{(\mu,v)}, \tag{41}$$

where the variable coefficients are given by

$$\alpha_{1,3,r}^{(1,\mu,v)} = Pr, \quad \alpha_{1,2,r}^{(1,\mu,v)} = \frac{\phi_1(10-\xi)}{15}f_r^{(\mu,v)} + \frac{\phi_1\xi(1-\xi)}{3}\frac{\partial f_r^{(\mu,v)}}{\partial \xi}, \quad \alpha_{1,0,r}^{(1,\mu,v)} = \frac{\phi_1(10-\xi)}{15}\frac{\partial^2 f_r^{(\mu,v)}}{\partial \eta^2},$$

$$\alpha_{1,1,r}^{(1,\mu,v)} = -\frac{\phi_1(10-4\xi)}{15}\frac{\partial f_r^{(\mu,v)}}{\partial \eta} - M^2\phi_1\phi_2 - \frac{\phi_1\xi(1-\xi)}{3}\frac{\partial}{\partial \xi}\left(\frac{\partial f_r^{(\mu,v)}}{\partial \eta}\right),$$

$$\alpha_{2,1,r}^{(1,\mu,v)} = -\frac{(1-\phi)^{2.5}}{15}(1+Pr)(10-4\xi), \quad \frac{(1-\phi)^{2.5}}{15}(1+Pr)(5+\xi)\eta, \quad \alpha_{2,1,r}^{(2,\mu,v)} = 1, \quad \alpha_{3,0,r}^{(2,\mu,v)} = -1,$$

$$\alpha_{3,2,r}^{(3,\mu,v)} = 1 + \left(1+\frac{k_f}{k_{nf}}Rd\right), \quad \alpha_{3,1,r}^{(3,\mu,v)} = \frac{\phi_4(10-\xi)}{15}f_r^{(\mu,v)} + \frac{\phi_4\xi(1-\xi)}{3}\frac{\partial f_r^{(\mu,v)}}{\partial \xi}, \quad \alpha_{3,0,r}^{(3,\mu,v)} = \frac{k_f}{k_{nf}}Q\xi$$

$$-\frac{\phi_4(1-\xi)}{3}\frac{\partial f_r^{(\mu,v)}}{\partial \eta}, \quad \alpha_{1,1,r}^{(3,\mu,v)} = -\frac{\phi_4(1-\xi)}{3}\theta_r^{(\mu,v)} - \frac{\phi_4\xi(1-\xi)}{3}\frac{\partial \theta_r^{(\mu,v)}}{\partial \xi}, \quad \alpha_{1,0,r}^{(3,\mu,v)} = \frac{\phi_4(10-\xi)}{15}\frac{\partial \theta_r^{(\mu,v)}}{\partial \eta},$$

$$\gamma_{1,1,r}^{(1,\mu,v)} = -\frac{\phi_1\xi(1-\xi)}{3}\frac{\partial f_r^{(\mu,v)}}{\partial \eta}, \quad \beta_{1,1,r}^{(1,\mu,v)} = \frac{\phi_1\xi(1-\xi)}{3}\frac{\partial^2 f_r^{(\mu,v)}}{\partial \eta^2}, \beta_{2,1,r}^{(1,\mu,v)} = -\frac{\phi_1\xi(1-\xi)}{3}(1+Pr),$$

$$\beta_{3,1,r}^{(3,\mu,v)} = -\frac{\phi_4\xi(1-\xi)}{3}\frac{\partial f_r^{(\mu,v)}}{\partial \eta}, \beta_{1,1,r}^{(3,\mu,v)} = \frac{\phi_4\xi(1-\xi)}{5}\frac{\partial \theta_r^{(\mu,v)}}{\partial \eta}, \quad R_{1,r}^{(\mu,v)} = \frac{\phi_1(10-\xi)}{15}f_r^{(\mu,v)}\frac{\partial^2 f_r^{(\mu,v)}}{\partial \eta^2}$$

$$-\frac{\phi_1(5-2\xi)}{15}\left(\frac{\partial f_r^{(\mu,v)}}{\partial \eta}\right)^2 - \frac{\phi_1\xi(1-\xi)}{3}\frac{\partial f_r^{(\mu,v)}}{\partial \eta}\frac{\partial}{\partial \xi}\left(\frac{\partial f_r^{(\mu,v)}}{\partial \eta}\right) + \frac{\phi_1\xi(1-\xi)}{3}\frac{\partial^2 f_r^{(\mu,v)}}{\partial \eta^2}\frac{\partial f_r^{(\mu,v)}}{\partial \xi},$$

$$R_{2,r}^{(\mu,v)} = 0, \quad R_{3,r}^{(\mu,v)} = \frac{\phi_4(10-\xi)}{15}f_r^{(\mu,v)}\frac{\partial \theta_r^{(\mu,v)}}{\partial \eta} - \frac{\phi_4(1-\xi)}{3}\theta_r^{(\mu,v)}\frac{\partial f_r^{(\mu,v)}}{\partial \eta}$$

$$-\frac{\phi_4\xi(1-\xi)}{5}\frac{\partial f_r^{(\mu,v)}}{\partial \eta}\frac{\partial \theta_r^{(\mu,v)}}{\partial \xi} + \frac{\phi_4\xi(1-\xi)}{5}\frac{\partial \theta_r^{(\mu,v)}}{\partial \eta}\frac{\partial f_r^{(\mu,v)}}{\partial \xi},$$

subject to boundary conditions

$$f_{r+1}^{(\mu,v)}(\xi,0) = 0, \quad \frac{\partial f_{r+1}^{(\mu,v)}}{\partial \eta}(\xi,0) = 0, \quad \xi\theta_{r+1}^{(\mu,v)}(\xi,0) - (1-\xi)^{6/5}\frac{\partial \theta_{r+1}^{(\mu,v)}}{\partial \eta}(\xi,0) = 1,$$

$$\frac{\partial f_{r+1}^{(\mu,v)}}{\partial \eta}(\xi,\infty) = 0, \quad \omega_{r+1}^{(\mu,v)}(\xi,\infty) = 0 \quad \theta_{r+1}^{(\mu,v)}(\xi,\infty) = 0. \tag{42}$$

We apply the Chebyshev spectral collocation method that uses bivariate Lagrange interpolation polynomials as basic functions as in the vertical plate. Thus, evaluating Equations (39) and (41) at the collocation points and making use of the derivative matrices as well as incorporating the initial conditions which corresponds to $\xi_{N_\xi}$, we obtain

$$A_{1,1}^{(\mu,v)}\mathbf{F}_{i,r+1}^{(\mu,v)} + A_{1,2}^{(\mu,v)}\mathbf{\Omega}_{i,r+1}^{(\mu,v)} + A_{1,3}^{(\mu,v)}\mathbf{\Theta}_{i,r+1}^{(\mu,v)} + \gamma_{1,r}^{(1,1,\mu,v)}\sum_{j=0}^{N_\xi}d_{i,j}\mathbf{D}^{(\mu)}\mathbf{F}_j^{(\mu,v)} + \beta_{1,1,r}^{(1,\mu,v)}\sum_{j=0}^{N_\xi}d_{i,j}\mathbf{F}_j^{(\mu,v)},$$

$$+\beta_{2,1,r}^{(1,\mu,v)}\sum_{j=0}^{N_\xi}d_{i,j}\mathbf{\Omega}_j^{(\mu,v)} = \mathbf{R}_{1,i}^{(\mu,v)}, \tag{43}$$

$$A_{2,1}^{(\mu,v)}\mathbf{F}_{i,r+1}^{(\mu,v)} + A_{2,2}^{(\mu,v)}\mathbf{\Omega}_{i,r+1}^{(\mu,v)} + A_{2,3}^{(\mu,v)}\mathbf{\Theta}_{i,r+1}^{(\mu,v)} = \mathbf{R}_{2,i}^{(\mu,v)}, \tag{44}$$

$$A_{3,1}^{(\mu,v)}\mathbf{F}_{i,r+1}^{(\mu,v)} + A_{3,2}^{(\mu,v)}\mathbf{\Omega}_{i,r+1}^{(\mu,v)} + A_{3,3}^{(\mu,v)}\mathbf{\Theta}_{i,r+1}^{(\mu,v)} + \beta_{1,1,r}^{(3,\mu,v)}\sum_{j=0}^{N_\xi}d_{i,j}\mathbf{F}_j^{(\mu,v)} + \beta_{3,1,r}^{(3,\mu,v)}\sum_{j=0}^{N_\xi}d_{i,j}\mathbf{\Theta}_j^{(\mu,v)} = \mathbf{R}_{3,i}^{(\mu,v)}. \tag{45}$$

The vectors $\mathbf{F}_{i,r+1}^{(\mu,v)}, \mathbf{\Omega}_{i,r+1}^{(\mu,v)}$ and $\mathbf{\Theta}_{i,r+1}^{(\mu,v)}$ denote the values of $f, \omega$ and $\theta$ approximated at the collocation points. Imposing boundary conditions for $i = 0, 1, 2, 3, \ldots, N_\xi - 1$, Equations (43)–(45) can be expressed as a matrix system of size $N_\xi(M+1) \times N_\xi(M+1)$ as in the previous subsection.

## 4. Results and Discussion

The transformed nonlinear PDEs for the vertical and horizontal plates were solved numerically using the OMD-BSQLM for Cu–water and Ag–water nanofluids. Numerical computations are carried out using $Pr = 0.7$ [9], $M = 0.5$ and $Q = 0.01$ [12,16]. However, the parametric values of the radiation parameter and nanoparticle volume fraction were chosen as $Rd = 0.6$ and $\phi = 0.3$. All of these values are treated the same in the entire study except the varied values in respective figures. The space domain $\eta$ was truncated to $\eta_\infty = 15$. The numerical results were generated using $N_\xi = 5$, $N_\eta = 20$ collocation points. The number of sub-intervals in both space and time are taken as $p = q = 5$. In order to obtain a clear understanding of the physics of the problem, a parametric study was undertaken to determine the impact of the different physical parameters on the fluid properties and flow characteristics.

To determine the accuracy of our numerical results, the local skin friction coefficient and the surface temperature are compared with the non-overlapping MD-BSQLM and published results by Yi and Lin [9] in Table 2. The table gives a comparison of the OMD-BSQLM results when $\xi = M = \phi = 0$ for different values of the Prandtl number $Pr$. It is observed that, for increasing values of the Prandtl number, the results are in good agreement with values in the literature and those obtained using the non-overlapping MD-BSQLM. Hence, the use of the present method is justified. It is also noted that the OMD-BSQLM can give accurate results with a minimal number of grid points compared to the non-overlapping MD-BSQLM. Table 3 presents results for the local skin friction $f''(\xi, 0)$, surface temperature $\theta(\xi, 0)$ and heat transfer rate $-\theta'(\xi, 0)$ for varying values of the dimensionless streamwise coordinate $\xi$ and different nanofluids. The table shows clearly that the skin friction, interfacial temperature, and heat transfer rate decrease with increasing values of $\xi$. This is due to the increase of the momentum boundary layer thickness and thermal boundary layer thickness.

**Table 2.** Comparison of the OMD-BSQLM results with MD-BSQLM, Yi and Lin [9] for $f''(0,0)$ and $\theta(0,0)$ at different values of $Pr$ when $\xi = M = \phi = 0$.

| | $\eta_\infty$ | $Pr$ | Yi and Lin [9] $f''(0,0)$ | $\theta(0,0)$ | MD-BSQLM $f''(0,0)$ | $\theta(0,0)$ | $N_\eta$ | OMD-BSQLM $f''(0,0)$ | $\theta(0,0)$ | $N_\eta$ |
|---|---|---|---|---|---|---|---|---|---|---|
| **Vertical plate** | | | | | | | | | | |
| | 12 | 0.001 | 54.745 | 1.3345 | 54.7463521 | 1.3344356 | 100 | 54.7463521 | 1.3344356 | 20 |
| | 12 | 0.01 | 16.929 | 1.3759 | 16.9295516 | 1.3758562 | 100 | 16.9295516 | 1.3758562 | 20 |
| | 12 | 0.1 | 5.2502 | 1.4824 | 1.2502342 | 1.4823999 | 100 | 1.2502342 | 1.4823999 | 20 |
| | 15 | 0.7 | 2.3123 | 1.6132 | 2.3123480 | 1.6129166 | 100 | 2.3123480 | 1.6129166 | 20 |
| | 15 | 7 | 1.5748 | 1.6520 | 1.5743519 | 1.6518940 | 100 | 1.5743519 | 1.6518940 | 20 |
| **Horizontal plate** | | | | | | | | | | |
| | 12 | 0.001 | 47.166 | 1.2258 | 47.2048673 | 1.2257703 | 100 | 47.2048673 | 1.2257703 | 20 |
| | 12 | 0.01 | 14.549 | 1.2720 | 14.5501264 | 1.2720149 | 100 | 14.5501264 | 1.2720149 | 20 |
| | 12 | 0.1 | 4.5424 | 1.3944 | 4.5423369 | 1.3943724 | 100 | 4.5423369 | 1.3943724 | 20 |
| | 15 | 0.7 | 2.0205 | 1.5583 | 2.0757356 | 1.5530446 | 100 | 2.0757356 | 1.5530446 | 20 |
| | 15 | 7 | 1.3622 | 1.6410 | 1.3618515 | 1.6413464 | 100 | 1.3618515 | 1.6413464 | 20 |

**Table 3.** OMD-BSQLM results for the skin friction coefficient, heat transfer rate and surface temperature at different values of $\xi$ when $Pr = 0.7, \phi = 0.3, M = 0.5, Q = 0.01$ and $Rd = 0.6$.

| | | | | | | |
|---|---|---|---|---|---|---|
| **Vertical Plate** | | | | | | |
| | **Cu-Water Nanofluid** | | | **Ag-Water Nanofluid** | | |
| $\xi$ | $f''(\xi,0)$ | $-\theta'(\xi,0)$ | $\theta(\xi,0)$ | $f''(\xi,0)$ | $-\theta'(\xi,0)$ | $\theta(\xi,0)$ |
| 0.1 | 3.1502197 | 0.8886284 | 2.2102538 | 3.4901209 | 0.8859706 | 2.2335515 |
| 0.2 | 2.9783142 | 0.7836947 | 2.0353093 | 3.2954705 | 0.7791175 | 2.0526246 |
| 0.3 | 2.8055275 | 0.6874567 | 1.8661087 | 3.1006493 | 0.6816661 | 1.8784674 |
| 0.4 | 2.6354158 | 0.6013814 | 1.7060758 | 2.9096575 | 0.5949833 | 1.7145224 |
| 0.5 | 2.4711001 | 0.5260086 | 1.5576812 | 2.7259292 | 0.5194660 | 1.5631829 |
| 0.6 | 2.3149074 | 0.4610286 | 1.4222383 | 2.5519493 | 0.4546541 | 1.4256179 |
| 0.7 | 2.1681887 | 0.4055214 | 1.2999488 | 2.3890815 | 0.3994990 | 1.3018590 |
| 0.8 | 2.0312907 | 0.3582374 | 1.1901080 | 2.2375714 | 0.3526578 | 1.1910409 |
| 0.9 | 1.9034739 | 0.3177938 | 1.0912546 | 2.0964877 | 0.3126913 | 1.0915734 |
| 1 | 1.7808520 | 0.2824192 | 1.0000000 | 1.9615149 | 0.2778150 | 1.0000000 |
| **Horizontal plate** | | | | | | |
| 0.1 | 2.0929952 | 0.8768285 | 2.2730937 | 2.1709163 | 0.8703743 | 2.3299696 |
| 0.2 | 1.9147577 | 0.7603163 | 2.0914784 | 1.9651128 | 0.7490266 | 2.1346661 |
| 0.3 | 1.7431456 | 0.6537272 | 1.9129913 | 1.7696198 | 0.6393135 | 1.9443078 |
| 0.4 | 1.5857671 | 0.5595096 | 1.7422446 | 1.5936813 | 0.5435780 | 1.7638210 |
| 0.5 | 1.4492674 | 0.4788634 | 1.5831252 | 1.4450572 | 0.4627300 | 1.5971702 |
| 0.6 | 1.3381426 | 0.4116324 | 1.4381961 | 1.3286458 | 0.3962184 | 1.4467514 |
| 0.7 | 1.2542162 | 0.3565745 | 1.3084563 | 1.2459879 | 0.3424128 | 1.3132268 |
| 0.8 | 1.1970874 | 0.3118610 | 1.1934924 | 1.1960098 | 0.2991830 | 1.1957896 |
| 0.9 | 1.1656347 | 0.2755951 | 1.0917901 | 1.1770321 | 0.2644615 | 1.0925707 |
| 1 | 1.1674280 | 0.2464322 | 1.0000000 | 1.2005270 | 0.2370678 | 1.0000000 |

Figures 4–7 depict the effects of nanoparticle volume fraction, thermal radiation, heat generation and magnetic field parameter on the velocity profiles for both Ag and Cu nanofluids. It is observed from the figures that Ag–water nanofluid shows better enhancement in the velocity profiles than Cu–water nanofluid. This is because the viscosity of the Ag–water nanofluid is higher compared to that of Cu–water nanofluid. The effect of using different types of nanofluids is more significant in the vertical plate than on the horizontal plate. Figure 4 shows the influence of the magnetic parameter on the dimensionless velocity. It is noted that the velocity is higher near the wall and lower far from the wall for hydrodynamic flows ($M = 0$). The opposite trend is observed for the hydromagnetic flows ($M \neq 0$). Moreover, increasing the magnetic parameter reduces the velocity distribution near the wall. The magnetic parameter is known to represent the Lorentz force that opposes the flow. The peak velocity decreases with the increasing values of the magnetic parameter due to the retarding effect in the boundary layer region. As a result, the separation of the boundary layer occurs earlier since the momentum boundary layer becomes thick. These findings concur with results reported by Mamun et al. [12] and Azim et al. [16] in regular fluids.

Figure 5 shows the effect of nanoparticle volume fraction on the velocity profiles. It is seen that the flow velocity increases around the vertical and horizontal plates with an increase in the nanoparticle volume fraction. For both the vertical and horizontal plates, it is clear that the flow velocity is significantly low for the conventional fluid ($\phi = 0$) than for nanofluids ($\phi \neq 0$). As expected, for the conventional fluid, there is no change in velocity profiles for both plates. However, as the volume fraction of nanoparticles increases, the velocity distribution also increases. This is due to an increase in the momentum boundary layer thickness which is attributed to adding nanoparticles to the base fluid. The nanoparticles enhance the velocity profiles due to the higher thermal conductivity of nanofluids. For the horizontal plate, we also observe that, near the wall, the momentum boundary layer thickness decreases as the volume fraction of silver particles increases and away from the wall, the boundary layer thickness increases. Figure 6 is presented to show the effect of the heat generation parameter on the dimensionless velocity. It is observed that more heat is generated within the boundary as the heat generation parameter increases and, consequently, the fluid velocity increases as well. The increase in

velocity is consistent with the physical consequence as the internal energy generation resulted from the heat generation increases the buoyancy forces, which in turn enhance more flow along both the vertical and horizontal plates. The effect of the thermal radiation parameter on the velocity distribution is shown in Figure 7 for both Ag–water and Cu–water nanofluids. We observe that the velocity increases within the boundary layer thickness as the thermal radiation parameter increases. Radiation accelerates the fluid motion, thus enhancing the velocity of nanofluids.

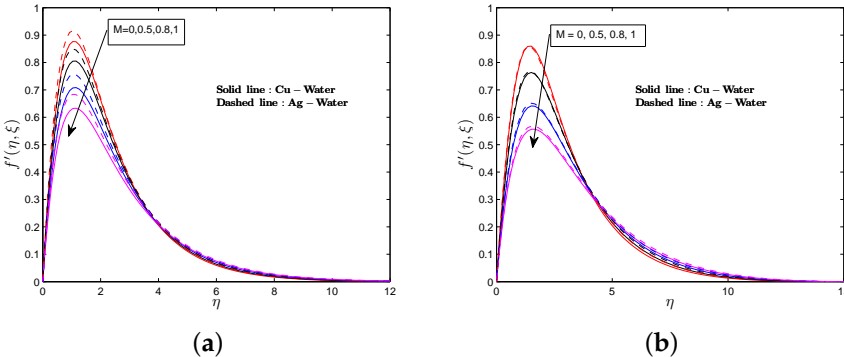

**Figure 4.** Velocity profiles for various values *M*. (**a**) vertical plate; (**b**) horizontal plate.

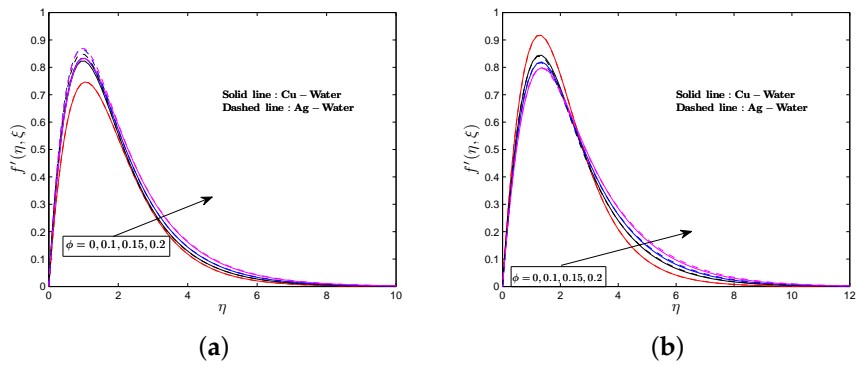

**Figure 5.** Velocity profiles for various values $\phi$. (**a**) vertical plate; (**b**) horizontal plate.

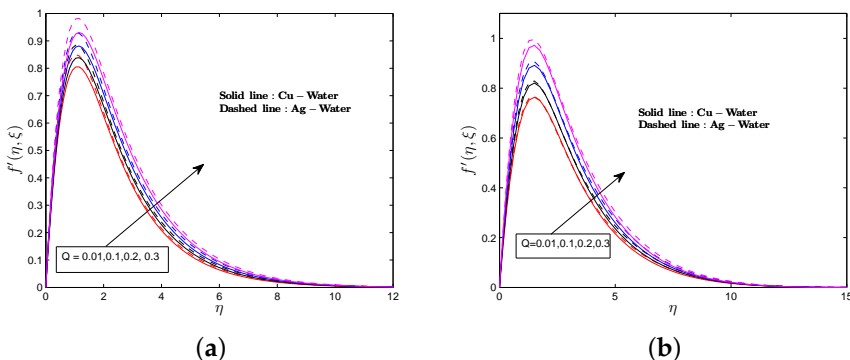

**Figure 6.** Velocity profiles for various values *Q*. (**a**) vertical plate; (**b**) horizontal plate.

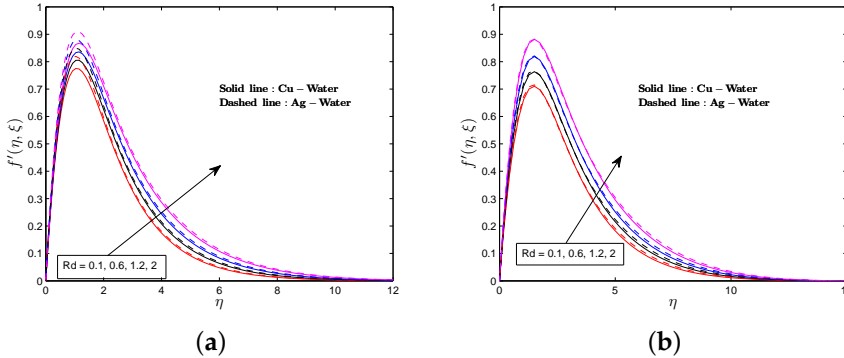

**Figure 7.** Velocity profiles for various values *Rd*. (**a**) vertical plate; (**b**) horizontal plate.

Figures 8–11 show the influence of thermal radiation, nanoparticle volume fraction, heat generation and magnetic field parameter on the temperature profiles for both nanofluids. The temperature distribution in the case of Ag–water nanofluid is relatively higher than in the case of Cu–water nanofluid. This is because the thermal conductivity of silver nanoparticles is higher than that of Copper nanoparticles. The effect of using different types of nanofluids is more clear on the horizontal plate than for the vertical plate. Figure 8 depicts the influence of the magnetic field parameter on the temperature profiles. The figure shows that the magnetic field enhances the thickness of the thermal boundary layer, thus increasing the temperature profiles. The effect of nanoparticle volume fraction on the temperature profiles is shown in Figure 9. For the vertical and horizontal plates, the thermal boundary layer thickness is enhanced when the nanoparticle volume fraction increases. Physically, increasing the nanoparticle volume fraction causes an increase in the thermal conductivity of the nanofluid, which in turn enhances the boundary layer thickness and an augmentation in the temperature profiles. Similar results were reported by Shahzad et al. [44]. It is worth mentioning that the temperature is significantly higher in the case of nanofluids than in the regular fluid ($\phi = 0$). This is due to the presence of high conductive silver and copper nanoparticles.

Figure 10 illustrates the effect of the heat generation parameter on the temperature profiles. It is seen that the thermal boundary layer is enhanced when the heat generation parameter increases. The energy resulted from internal heat generation increases the temperature of the fluid within the boundary and increases the motion of the fluid. The influence of thermal radiation on the temperature profiles is shown in Figure 11. The figure depicts that an increase in the thermal radiation parameter improves the temperature profiles. As the temperature increases with increasing radiation parameter, the thickness of the thermal boundary layer is enhanced. The larger values of the amount of $\frac{k^* k_{nf}}{4\sigma^* T_\infty^3}$ in the radiation parameter indicate dominance in the thermal radiation over conduction. Thus, there is a large amount of radiative heat energy being poured into the system. The fluid within the boundary layer absorbs imitated heat from the heated plate because of the radiation effect. The radiated heat ultimately increases the temperature of the fluid. The greater values of thermal radiation parameter generate higher temperature and, consequently, the fluid motion is accelerated. Similar results were obtained by Ali et al. [32] in the case of regular fluid.

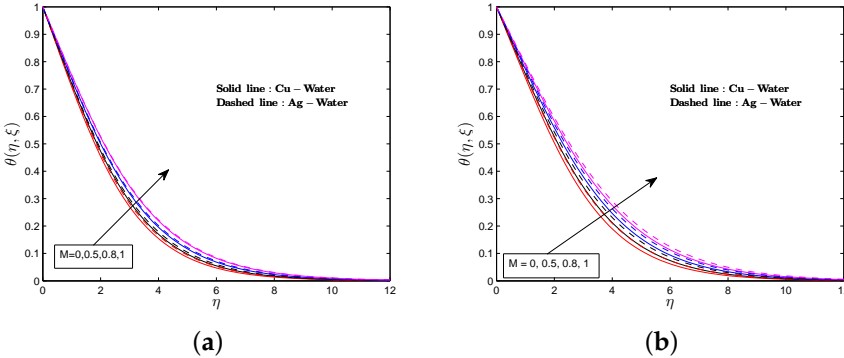

**Figure 8.** Temperature profiles for various values of *M*. (**a**) vertical plate; (**b**) horizontal plate.

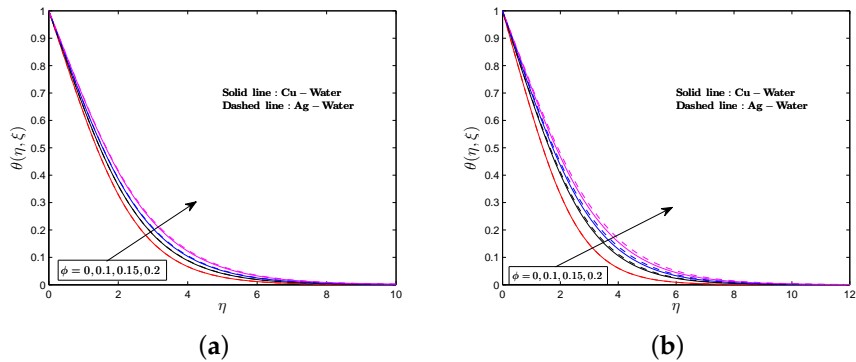

**Figure 9.** Temperature profiles for various values *ϕ*. (**a**) vertical plate; (**b**) horizontal plate.

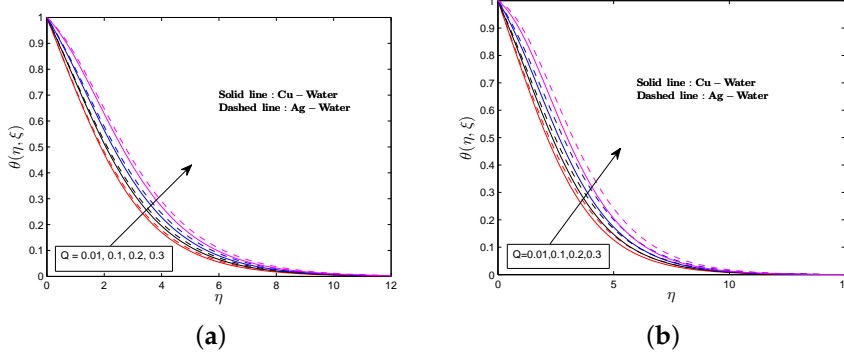

**Figure 10.** Temperature profiles for various values of *Q*. (**a**) vertical plate; (**b**) horizontal plate.

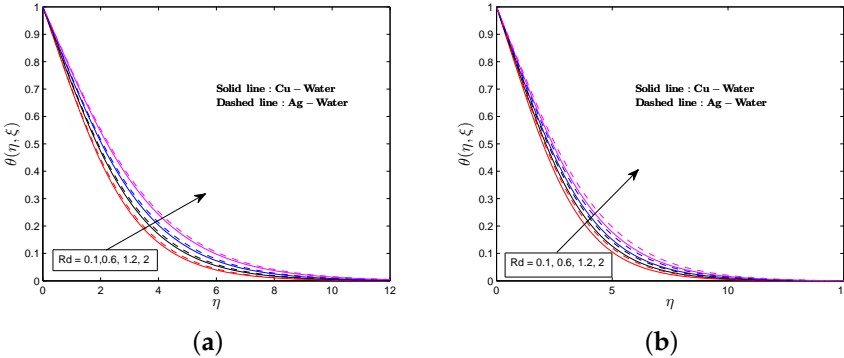

**Figure 11.** Temperature profiles for various values *Rd*. (**a**) vertical plate; (**b**) horizontal plate.

Figures 12–15 present the variation of the local skin friction at different values of the thermal radiation, heat generation, nanoparticle volume fraction and magnetic parameter for the Ag–water and Cu–water nanofluids. The skin friction coefficient is higher in the Ag–water nanofluid compared to the Cu–water nanofluid. Hence, the Ag–water nanofluid gives a high drag force in opposition to the flow compared to the Cu–water nanofluid. For the vertical plate, the skin friction is higher for the Ag–water nanofluid throughout the surface. However, for the horizontal plate, the skin friction is higher for a Ag–water nanofluid close to the wall and higher for a Cu–water nanofluid far from the wall. The effect of the magnetic parameter on the skin friction coefficient is shown in Figure 12. The figure shows that, when the magnetic parameter increases, the skin friction coefficient decreases. The magnetic force that opposes the flow decreases the shear stress at the wall, thus reducing the skin friction coefficient.

The behaviour of the skin friction coefficient against the streamwise coordinate $\xi$ for different values of the nanoparticle volume fraction is plotted in Figure 13. The figure shows that an increase in the nanoparticle volume fraction causes a decrease in the skin friction at the plates. In Figure 14, the impact of the heat generation parameter on the local skin friction is exhibited. The figure reflects that the skin friction factor increases with increasing heat generation parameter. As mentioned earlier, increasing the heat generation parameter accelerates the flow and generates greater buoyancy force and thus increases the skin friction coefficient. The influence of the thermal radiation parameter on the skin friction coefficient is shown in Figure 15. The increase in the fluid motion due to thermal radiation enhances the shear stress at the wall which in turn causes an increase in the skin friction coefficient.

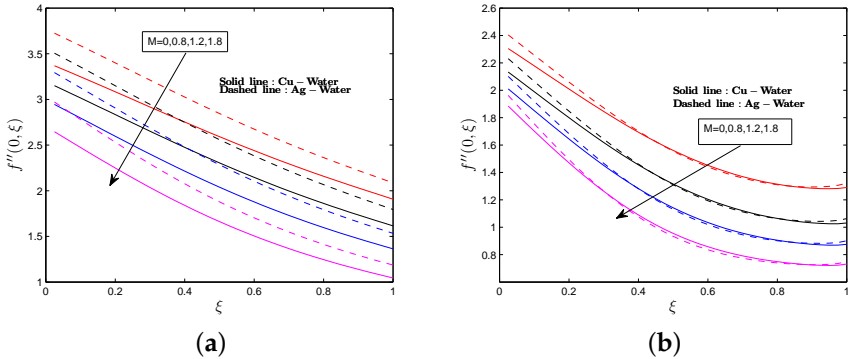

**Figure 12.** Skin friction coefficient for various values of *M*. (**a**) vertical plate; (**b**) horizontal plate.

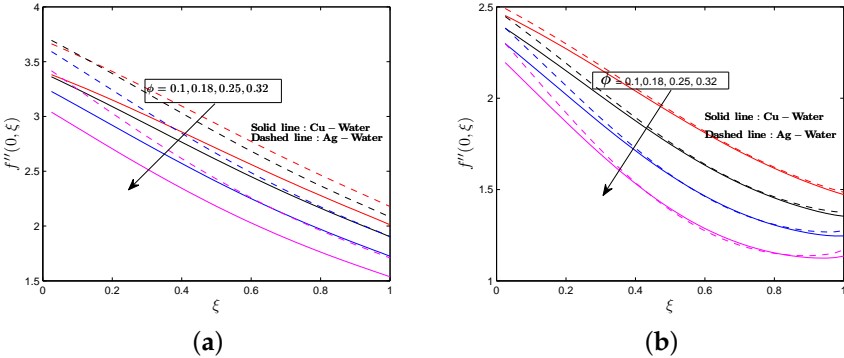

**Figure 13.** Skin friction coefficient for various values of $\phi$. (**a**) vertical plate; (**b**) horizontal plate.

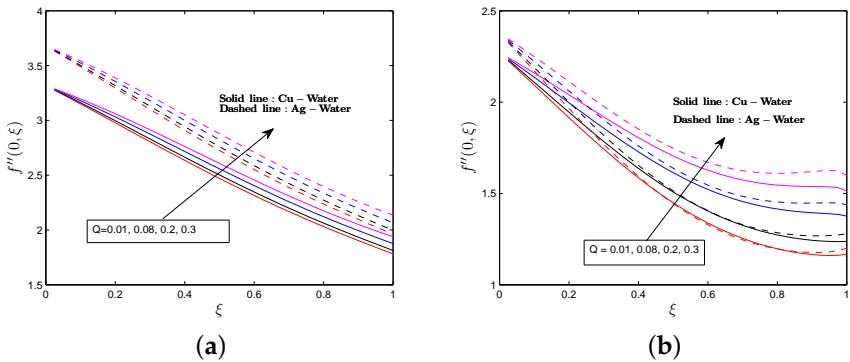

**Figure 14.** Skin friction for various values of $Q$. (**a**) vertical plate; (**b**) horizontal plate.

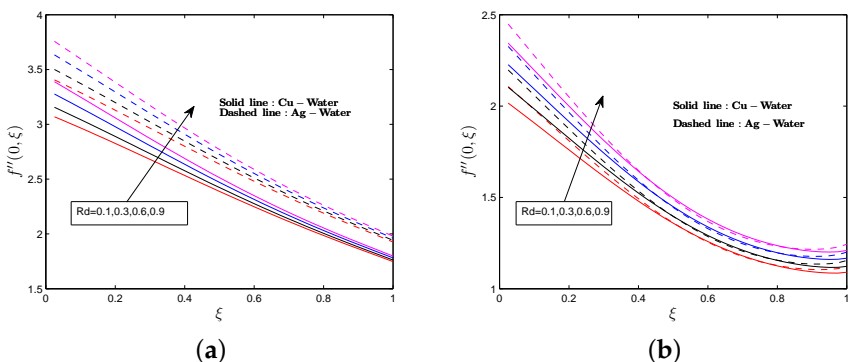

**Figure 15.** Skin friction coefficient for various values of $Rd$. (**a**) vertical plate; (**b**) horizontal plate.

Figures 16–19 show the effects of the thermal radiation, heat generation, nanoparticle volume fraction and magnetic field parameter on the rate of heat transfer for both Ag and Cu nanofluids. The rate of heat transfer is observed to be higher in the Cu–water nanofluid than in the Ag–water nanofluid for both vertical and horizontal plates. An increase in the magnetic parameter reduces the rate of heat transfer as seen in Figure 16. The increasing magnetic field parameter enhances the thermal boundary layer thickness and consequently the heat transfer rate decreases due to an increase in the magnetic field strength. In addition, the rate of heat transfer depends on the gradient of temperature and, as the temperature gradient decreases with increasing values of the magnetic parameter, the heat transfer rate also decreases.

Figure 17 depicts the impact of the nanoparticle volume fraction on the skin friction for the different nanofluids. Increasing the nanoparticle volume fraction enhances the thermal conductivity of

the nanofluids, which reduces the thermal boundary layer thickness and the temperature gradient at the wall as observed from the figure. The influence of the heat generation on the heat transfer rate is depicted in Figure 18. The figure shows that the heat transfer rate decreases with increasing heat generation parameter. Since higher values of the heat generation parameter create a hot layer of fluid near the surface which results in the temperature of the fluid to exceed the surface temperature, accordingly, the rate of heat transfer from the surface decreases. Figure 19 depicts that increasing values of the thermal radiation parameter enhances the fluid interfacial temperature, which in turn makes the flow of the heat rate slower.

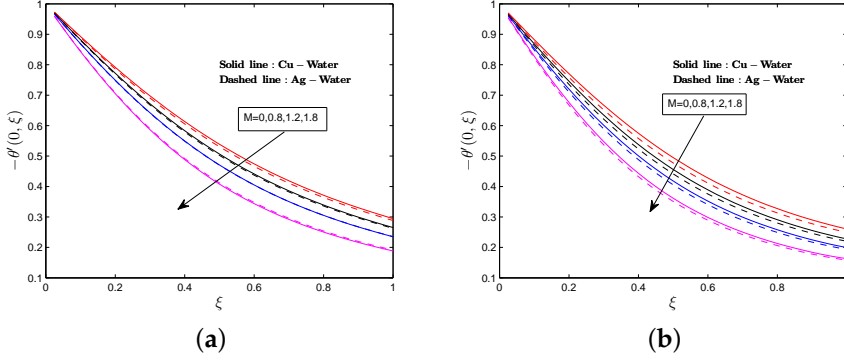

**Figure 16.** Heat transfer rate for various values of *M*. (**a**) vertical plate; (**b**) horizontal plate.

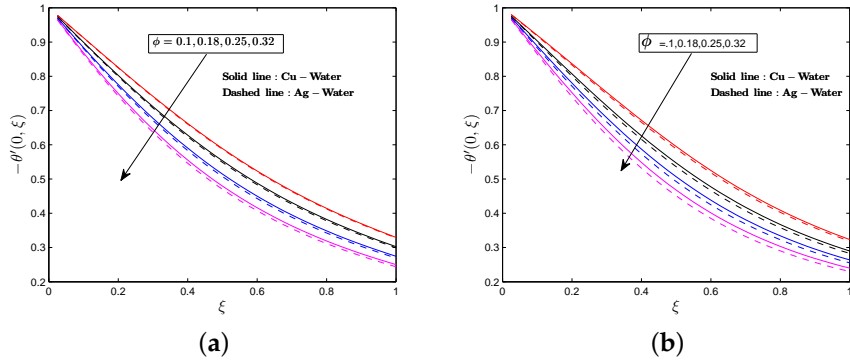

**Figure 17.** Heat transfer rate for various values of $\phi$. (**a**) vertical plate; (**b**) horizontal plate.

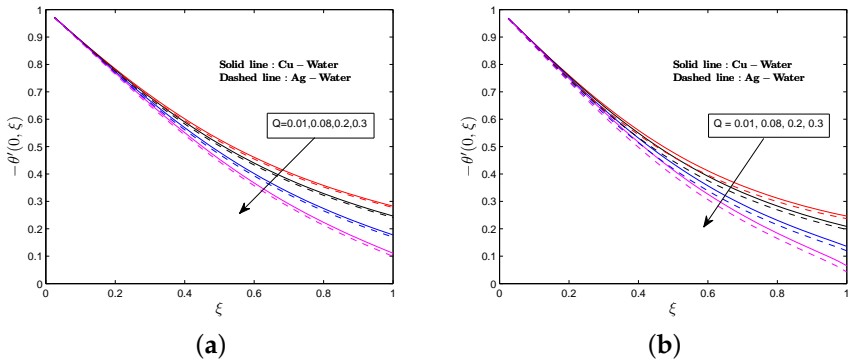

**Figure 18.** Heat transfer rate for various values of *Q*. (**a**) vertical plate; (**b**) horizontal plate.

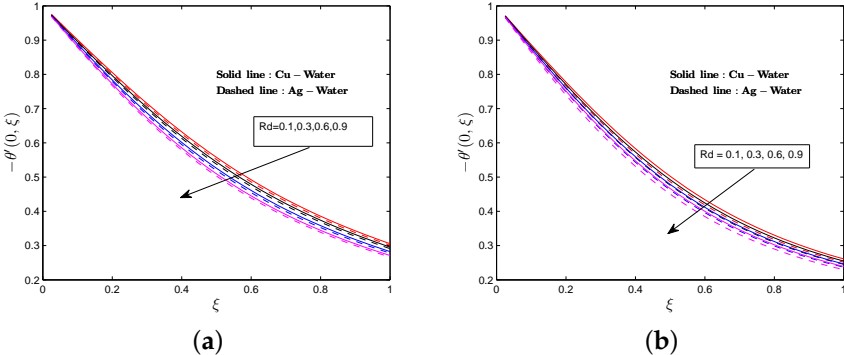

**Figure 19.** Heat transfer rate for various values of *Rd*. (**a**) vertical plate; (**b**) horizontal plate.

Figures 20–23 display the influence of the thermal radiation, heat generation, nanoparticle volume fraction and magnetic field parameter on the surface temperature for the Ag–water and Cu–water nanofluids. The surface temperature is noted to be higher in the Ag–water nanofluid than in the Cu–water nanofluid for both the vertical and horizontal plates. In Figure 20, we observe that, when the magnetic field is applied in the system, the surface temperature is enhanced for both the vertical and horizontal plates. As the magnetic field increases, the surface temperature is enhanced. The interaction between the magnetic field and the fluid motion increases the temperature of the fluid within the boundary layer which in turn increases the thermal boundary layer thickness as well as the surface temperature. Figure 21 shows that adding nanoparticles to the fluid enhances the surface temperature since the surface temperature increases with increasing nanoparticle volume fraction. The surface temperature increases with increasing values of the heat generation parameter as observed in Figure 22. This is because the temperature within the boundary layer increases for increasing heat generation parameter and thus enhances the surface temperature. Figure 23 shows that increasing the thermal radiation parameter also enhances the surface temperature.

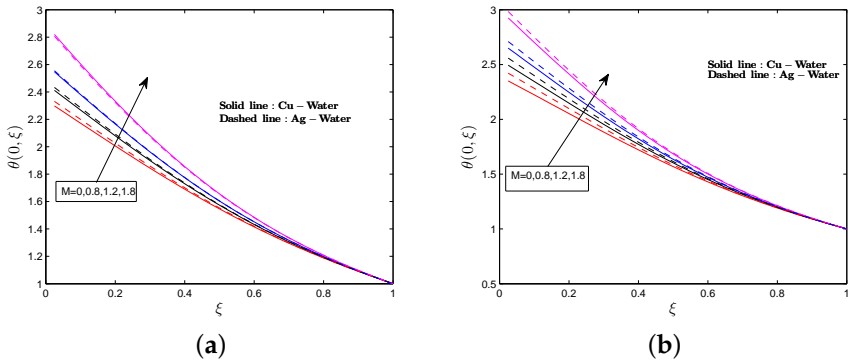

**Figure 20.** Surface temperature for various values of *M*. (**a**) vertical plate; (**b**) horizontal plate.

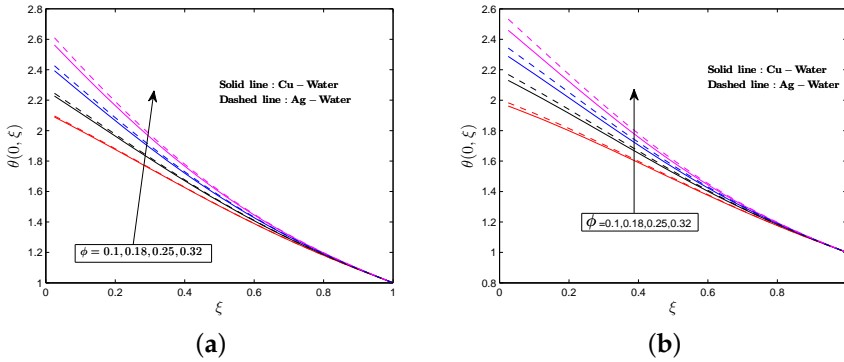

**Figure 21.** Surface temperature for various values of $\phi$. (**a**) vertical plate; (**b**) horizontal plate.

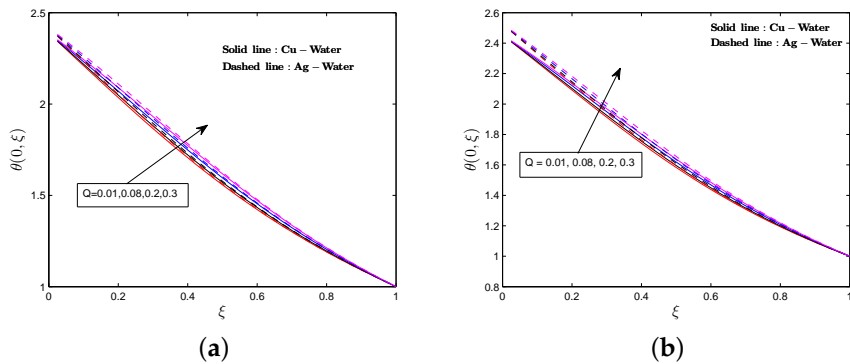

**Figure 22.** Surface temperature for various values of $Q$. (**a**) vertical plate; (**b**) horizontal plate.

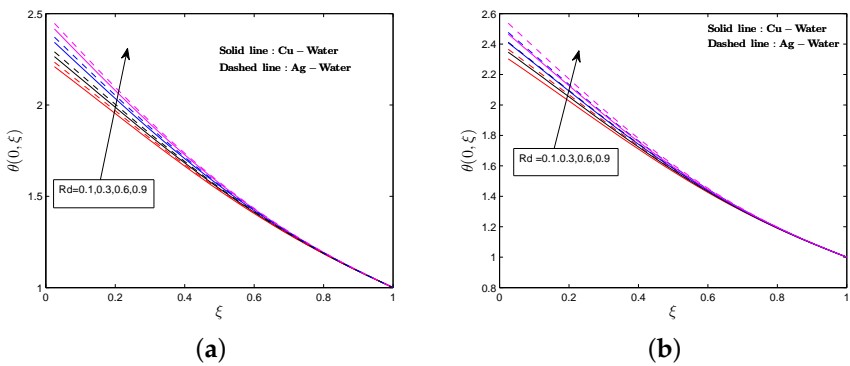

**Figure 23.** Surface temperature for various values of $Rd$. (**a**) vertical plate; (**b**) horizontal plate.

## 5. Conclusions

The multi-domain bivariate spectral quasilinearisation method was used to analyze conjugate heat transfer in MHD free convection flow of copper water and silver water nanofluids over vertical and horizontal plates. The comparison with previously published results was performed and the results were in good agreement. The effects of nanofluids, heat generation, thermal radiation, nanoparticle volume fraction and magnetic field parameter on the fluid properties and flow characteristics were discussed appropriately with numerical computations. The results obtained in the present study can have practical importance in various problems such as ablation or perspiration cooling problems. The results of the paper are of engineering interest where heat transfer processes are controlled in

polymer processing and nuclear reactor cooling systems. and designing and operation of plate heat exchangers. From the obtained results and discussion, the following conclusions can be drawn:

- The Ag–water nanofluid has higher velocity and temperature profiles, skin friction coefficient, and surface temperature than the Cu–water nanofluid. However, the reverse is true for the rate of heat transfer.
- Heat generation, thermal radiation, nanoparticle volume fraction and magnetic field parameter enhance the velocity of the nanofluid far from the wall. However, an increase in the magnetic field parameter significantly decreases the velocity of the nanofluid near the wall.
- Increasing the heat generation, thermal radiation, nanoparticle volume fraction and magnetic field parameter improves the temperature distribution and the surface temperature while reducing the rate of heat transfer.
- The overlapping multi-domain bivariate spectral quasilinearisation method holds great potential for solving highly nonlinear conjugate heat transfer problems since the method gives accurate results using a minimal number of grid points.

**Author Contributions:** Conceptualization, M.M.; Data curation, M.M.; Methodology, M.M. and S.M.; Supervision, S.M. and P.S.; Validation, S.M. and P.S.; Writing—original draft, M.M.; Writing—review and editing, M.M., S.M. and P.S.

**Funding:** This research received no external funding.

**Acknowledgments:** The authors would like to thank the anonymous reviewers for their valuable comments and suggestions to improve the quality of the paper.

**Conflicts of Interest:** The authors declare no conflict of interest.

## Abbreviations

The following abbreviations are used in this manuscript:

| | |
|---|---|
| $B(x)$ | External uniform magnetic field |
| $B_0$ | Magnetic strength |
| $p$ | Pressure |
| $R_a$ | Rayleigh number |
| $g$ | Gravitational acceleration |
| $k$ | Thermal conductivity (W/m K) |
| $C_p$ | Specific heat capacity |
| $T$ | Fluid temperature (K or $^\circ$C ) |
| $q_h$ | Heat flux |
| $f$ | Dimensionless stream function |
| $(u, v)$ | Velocity component in Cartesian coordinate |
| $T_b$ | Constant temperature |
| $T_\infty$ | Ambient temperature |
| $Q_0$ | Rate of heat generation |
| $q_r$ | Radiative heat flux |
| $M$ | Magnetic field parameter |
| $Pr$ | Prandtl number |
| $Rd$ | Radiation parameter |
| $Q$ | Heat generation parameter |

**Greek Symbols**

| | |
|---|---|
| $\eta$ | Scaled boundary layer coordinate |
| $\zeta$ | Streamwise coordinate |
| $\sigma$ | Electrical conductivity (S m$^{-1}$) |
| $\alpha$ | Thermal diffusivity m$^2$s$^{-1}$ |
| $\mu$ | Dynamic viscosity kg m$^{-1}$s$^{-1}$ |
| $\theta$ | Dimensionless temperature |
| $\phi$ | Nanoparticle volume fraction parameter |
| $\psi$ | Stream function m$^2$s$^{-1}$ |
| $\rho$ | Density of the fluid ( Kg/m$^3$) |
| $\beta$ | Thermal expansion coefficient |
| $\nu$ | Kinematic viscosity m$^2$s$^{-1}$ |

**Subscripts**

| | |
|---|---|
| $nf$ | Nanofluid phase |
| $f$ | Fluid phase |
| $s$ | Solid phase |

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
