# Peer review of "Overlapping Multi-Domain Spectral Method for Conjugate Problems of Conduction and MHD Free Convection Flow of Nanofluids over Flat Plates"

_mca, doi:10.3390/mca24030075_

Reviewer 1 Report

Title: Overlapping Multi-Domain Spectral Method for Conjugate Problems of Conduction and MHD Free Convection Flow of Nanofluids over Flat Plates

Decision: major revision

This study presents an efficient overlapping multi-domain spectral method which used in the analysis of conjugate problems of heat conduction in solid walls coupled with laminar MHD free convective boundary layer flow. The computed numerical results are compared with previously published results and found to be in good agreement. Considering extensive efforts of this work, it can be considered for publication if the article is modified satisfactorily. The following suggestions are provided for further improving the quality of the manuscript.

1. The advantages of Overlapping Multi-Domain Spectral Method for Conjugate Problems should be explained deeply in introduction.

2. Academic words such as MHD that first appear in abstracts and texts should be given the full name, some main conclusions should be presented in the abstract.

3. In the introduction, the introduction of nanofluids should be mentioned separately, and some latest references should be reconsidered. (Such as Energy Conversion and Management, 2018, 171: 272-278. and Energy Conversion and Management, 2018, 177: 249-257)

4. According to the thermo-physical properties of nanofluids are researched in this paper, the nanofluid constants should be defined as Eq. (5) and add the reference.

5. The “B”, external uniform magnetic field, in Figure 1 should be explained and showed in equation. The spatial equation of the magnetic field should be described.

6. The multiphase flow in nanofluid composed of nanoparticles and base fluids should be considered, and the conjugate heat transfer is not described specifically by mathematical formulas.

7. The style of figure and reference should be further confirmed, in introduction, no recognizable source can be found according to the references provided

8. Conclusions should be summed up in two or three paragraphs summarizing your valuable results.

Reviewer 2 Report

The paper reports overlapping multi-domain spectral method for conjugate problems of conduction and MHD free convection flow of nanofluids over flat plates.

The Authors have discussed the topic in a very interesting way. The paper is written well, but I suggest to add information about another elements significantly different in properties. I would also like to ask the Authors to outline possible applications resulting from the facts described.

I think that the manuscript has sufficient scientific quality and relevance for Mathematical and Computational Applications. I suggest accept the publication as it is.

Convection Flow of                

Author Response

Round  2

Reviewer 1 Report

Accept in present form.